# BAR: Refactor the Basis of Autoregressive Visual Generation

**Zhicong Tang**[1]**, Dong Chen**[2]**, Jianmin Bao**[2]**, Baining Guo**[2]
[1]Tsinghua University, [2]Microsoft Research Asia

## Abstract

Autoregressive (AR) models, despite their remarkable successes, encounter limitations in image generation due to sequential prediction of tokens, *e.g.* local image patches, in a predetermined row-major raster-scan order. Prior works improve AR with various designs of prediction units and orders, however, rely on human inductive biases. This work proposes Basis Autoregressive (BAR), a novel paradigm that conceptualizes tokens as basis vectors within the image space and employs an end-to-end learnable approach to transform basis. By viewing tokens $x_k$ as the projection of image $\mathbf{x}$ onto basis vectors $e_k$, BAR's unified framework refactors fixed token sequences through the linear transform $\mathbf{y} = \mathbf{Ax}$, and encompasses previous methods as specific instances of matrix $\mathbf{A}$. Furthermore, BAR adaptively optimizes the transform matrix with an end-to-end AR objective, thereby discovering effective strategies beyond hand-crafted assumptions. Comprehensive experiments, notably achieving a state-of-the-art FID of 1.15 on the ImageNet-256 benchmark, demonstrate the ability of BAR to overcome human biases and significantly advance image generation, including text-to-image synthesis.

## 1 Introduction

Autoregressive (AR) models have demonstrated remarkable success in various domains (OpenAI, 2022; Team et al., 2023), particularly in natural language processing (Brown et al., 2020; Devlin et al., 2019; Raffel et al., 2020). This paradigm of sequentially predicting next tokens is extended to the vision domain with notable progress (Alayrac et al., 2022; Yu et al., 2022), including image generation (Esser et al., 2021; Ramesh et al., 2021; Han et al., 2024; Yu et al., 2024a), and even surpasses diffusion models (Song et al., 2021; Ho et al., 2020; Podell et al., 2024; Tang et al., 2022). However, prevalent AR models flatten images as 1D sequences of tokens in a row-major raster-scan order. While it aligns with the sequential nature of language, it overlooks the inherent 2D structure of images, where each token exhibits strong relationships with its neighbors. This straightforward adaptation significantly limits the capabilities and further development of AR models.

Recognizing these limitations, recent research (Fan et al., 2024; Li et al., 2024b; Yu et al., 2024c; Li et al., 2025; Pang et al., 2024) explores alternative strategies to suit the characteristics of images within the AR framework. VAR (Tian et al., 2024) moves from the standard next-token prediction to the coarse-to-fine next-scale prediction. MAR (Li et al., 2024b) transforms the traditionally causal, unidirectional generation into a bidirectional attention mechanism. Simultaneously, other explorations (Ren et al., 2025; Yu et al., 2024b; Wang et al., 2024; Yu et al., 2025) also investigate various aspects such as flexible token definitions, randomized generation orders, parallel processing, and frequency-based generation to address the inherent challenges of AR models.

However, these advancements are hindered by two fundamental challenges. First, they rely heavily on manual designs and inductive biases. VAR (Tian et al., 2024) is based on the coarse-to-fine causality inspired by human perception, FAR (Yu et al., 2025) opts to exploit frequency-domain hierarchy, while xAR (Ren et al., 2025) directly groups adjacent tokens as cells. These ad hoc choices yield divergent conclusions from their own inductive biases. Second, these approaches lack a unified mathematical framework and formal foundations, undermining their credibility and persuasiveness. For instance, PAR (Wang et al., 2024) partitions tokens by their locations, RAR (Yu et al., 2024b) randomly permutes tokens and gradually anneals to normal order, and xAR (Ren et al., 2025) empirically adopts cell as the basic entity.

To address these gaps, we propose Basis Autoregressive (BAR), including two pivotal contributions, *e.g.* a unified mathematical framework and a parameterized learnable algorithm. First, our framework is grounded in the theory of linear spaces and formalizes prior AR variants (Tian et al., 2024; Ren et al., 2025; Yu et al., 2024b; Wang et al., 2024; Yu et al., 2025) as specific transforms of space. They essentially re-mix, re-order, and re-group the token sequences with manually defined rules, while our framework is compatible with them and further offers a generalized viewpoint. Second, leveraging this framework, we introduce a parameterized, learnable, and end-to-end optimization algorithm that does not rely on heuristic designs and human biases. This avoids reliance on hand-crafted priors, eliminates extensive experimental trial and errors, and allows the model to adaptively discover optimal transforms through training.

As visualized in fig. 1, we first partition the latent space of token sequences $\mathbf{x}$ into a series of sub-spaces, where each token $x_i$ is the projection on them. Then, we apply the linear transform $\mathbf{y} = \mathbf{A}\mathbf{x}$, and the row vectors of $\mathbf{A}$ form the basis of transformed space. Prior methods (Tian et al., 2024; Ren et al., 2025; Yu et al., 2024b; Wang et al., 2024; Yu et al., 2025) are exemplified as certain forms of $\mathbf{A}$. Then, we propose a joint learning algorithm in fig. 2 for the matrix with the training objectives derived from existing AR models. Our method enables end-to-end training that seamlessly integrates with them and is firmly supported by comprehensive experiments on conditional and text-to-image generation, profound ablations of the learned transform matrix, and the state-of-the-art FID score of 1.15 on ImageNet 256 benchmark. Our contributions are:

- A unified framework that formalizes former AR methods and facilitates novel extensions;
- An end-to-end learnable algorithm that transcends human biases into adaptive optimization;
- Comprehensive experiments and ablations underscore the advantages of our method.

## 2 BACKGROUND

### 2.1 DISCRETE AUTOREGRESSIVE VISUAL GENERATION

Autoregressive (AR) models for visual generation adapt the paradigm of sequential modeling from the field of language, primarily through discrete tokenization. While early approaches (Van den Oord et al., 2016; Van Den Oord et al., 2016) directly model pixel-level dependencies, VQ-VAE (Van Den Oord et al., 2017; Razavi et al., 2019) introduces vector quantization and maps images to discrete tokens. Later works (Ramesh et al., 2021; Yu et al., 2022) scale to text-to-image generation, demonstrating the potential of AR. LlamaGen (Sun et al., 2024) further adapts large language model architectures, *e.g.* Llama (Touvron et al., 2023). Given sequential tokens $\mathbf{x} = \{x_1, x_2, \ldots, x_N\}$ representing an image, AR assumes that each token $x_k$ depends only on its prefix $x_{<k} := \{x_1, x_2, \ldots, x_{k-1}\}$, and factorizes the joint distribution $p_\theta(\mathbf{x})$ into a product of conditioned probabilities over the sequence as $p_\theta(\mathbf{x}) = \prod_{k=1}^{N} p_\theta(x_k \mid x_{<k})$.

Recent advancements focus on optimizing token quantization, prediction, and efficiency. RQ-Transformer (Lee et al., 2022) introduced residual quantization to reduce codebook redundancy, while TiTok (Yu et al., 2024c) reduces the number of required tokens to encode an image down to 32. VAR (Tian et al., 2024) pioneered a coarse-to-fine next-scale prediction paradigm, using a custom multi-scale RQ-VAE to generate tokens at increasing resolutions. PAR (Wang et al., 2024) introduces parallel decoding by identifying weakly dependent tokens as groups. RAR (Yu et al., 2024b) addresses fixed factorization orders via randomized permutations, gradually anneals to normal orders during training, and learns bidirectional contexts of images.

### 2.2 CONTINUOUS AUTOREGRESSIVE VISUAL GENERATION

Continuous AR models bypass vector quantization to directly model high-fidelity visual contents. Hybrid approaches (Zhou et al., 2024; Xie et al., 2024) attempt to bridge discrete and continuous visual data with multi-modal models. Later, MAR (Li et al., 2024b) eliminated vector quantization by integrating a diffusion loss, where the output tokens of AR are fed into tiny denoising networks as conditions to generate continuous outputs. Specifically, the output of AR models, noted as $z_k$, no longer matches the image tokens and serves as the condition for the noisy estimator $\epsilon_\eta$. The noisy estimator is trained via a denoising criterion

$$\mathcal{L}_{\text{MAR}}(z_k, x_k) = \left\| \epsilon - \epsilon_\eta(x_k^t | t, z_k) \right\|_2^2, \tag{1}$$

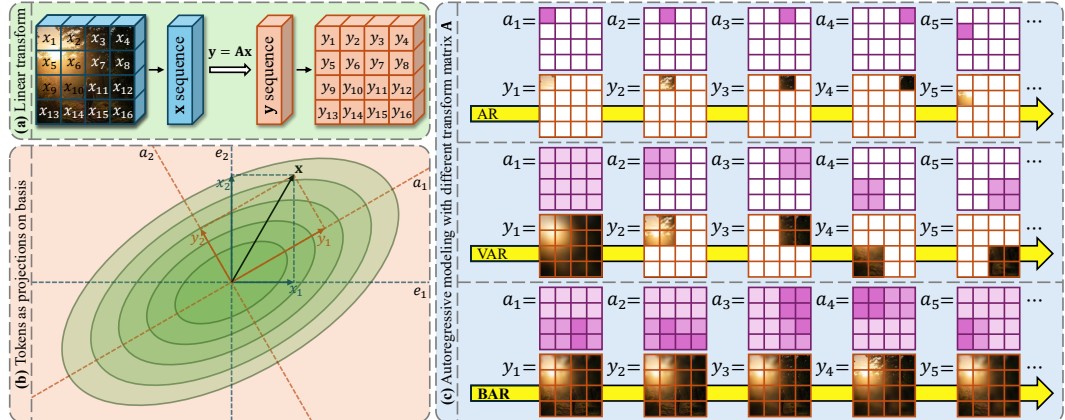

Figure 1: An overview of the unified framework of BAR and its strength over previous approaches. **(a)** By applying a linear transform associated with the matrix $\mathbf{A}$, BAR offers the generalized viewpoint that encompasses prior methods as specific instances of $\mathbf{A}$ and facilitates further extensions. **(b)** BAR at its core lies that each token $x_k$ is the projection of whole image $\mathbf{x}$ on a sub-space, or basis with channels omitted. It transforms the standard basis $e_k$ into the row vectors $a_k$ of matrix $\mathbf{A}$. **(c)** We illustrate each method with its corresponding $a_k$. While vanilla AR directly employs $e_i$ as raster scan of tokens and VAR manually designs a coarse-to-fine pattern, BAR adaptively learns $a_k$.

where $\epsilon \sim \mathcal{N}(0, 1)$ is random noise, $x_k^t = \sqrt{\bar{\alpha}_t} x_k + \sqrt{1 - \bar{\alpha}_t} \epsilon$ is the noise-corrupted sample, and $\bar{\alpha}$ is the noise schedule. Following works like FAR (Yu et al., 2025) in turn adopted a frequency-domain strategy, generating low-to-high frequency components to align with visual hierarchies, and captures spatial dependencies efficiently. Furthermore, xAR (Ren et al., 2025) eliminates the diffusion heads of MAR, drives the decoder to directly predict the continuous tokens, and uses groups of local tokens, *e.g.* cell, as the unit of each AR step. xAR also utilizes the flow-based (Lipman et al., 2023; Liu et al., 2023) objective

$$\mathcal{L}_{\text{xAR}}(\mathbf{x}) = \sum_{k=1}^{N} \left\| v_\theta(\{x_1^{t_1}, x_2^{t_2}, \ldots, x_k^{t_k}\}, t_k) - v_k^{t_k} \right\|_2^2, \tag{2}$$

where $x_k^{t_k} = (1 - t_k)x_k + t_k \epsilon_k$ is the noisy sample, $v_k^{t_k} = \frac{dx_k^{t_k}}{dt_k} = \epsilon_k - x_k$ is the ground-truth flow, $\{t\} = \{t_1, t_2, \ldots, t_n\} \sim \mathbf{U}[0, 1]$ are timesteps, and $\{\epsilon\} = \{\epsilon_1, \epsilon_2, \ldots, \epsilon_n\} \sim \mathcal{N}(0, 1)$ are noise.

## 3 METHOD

### 3.1 UNIFIED FRAMEWORK

Consider an image encoded into a 2D feature grid $\{x_{(i,j)}\}$, where each element corresponds to a local patch of image, $x_{(i,j)} \in \mathbb{R}^d$ in the case of VAE, and $x_{(i,j)} \in \mathbb{Z}_K := \{1, \ldots, K\}$ in the case of VQ-VAE. AR flattens it into a 1D sequence of tokens $\mathbf{x} = \{x_1, x_2, \ldots, x_N\}$. The entire image can also be viewed as a vector $\mathbf{x} \in \mathbb{R}^{N \times d}$ or $\mathbf{x} \in \mathbb{Z}_K^N$ for continuous and discrete cases, respectively. AR transforms the modeling of the whole image $\mathbf{x}$ into the progressive predictions of each token $x_k$, which can be viewed as the projection from the high-dimensional $\mathbf{x}$ yielding a series of low-dimensional $x_k$ in different sub-spaces. In the following discussion, we omit the channel dimension $d$ for simplicity, since the transform can be independently applied on each channel.

Specifically, consider the standard basis $\{e_1, e_2, \ldots, e_N\}$ of $\mathcal{S} := \mathbb{R}^N$, where $e_k = \text{onehot}(k) \in \mathbb{R}^N$ with its $k$-th element being one and others being zeros. AR splits the $\mathcal{S}$ into the sub-spaces

$$\{\mathcal{S}_k | \mathcal{S}_k := \text{span}(e_k), 1 \leq k \leq N\}, \tag{3}$$

where $\text{span}(e_k)$ represents the space spanned by $e_k$. AR transforms the direct modeling of $\mathbf{x}$ into progressively determining its projections onto $\mathcal{S}_k$, *e.g.* each token $x_k$. However, specific designs of these sub-spaces $\{\mathcal{S}_k\}$ remain understudied, as most AR models simply adopt the vanilla form above that corresponds to the row-major raster-scan of image patches.

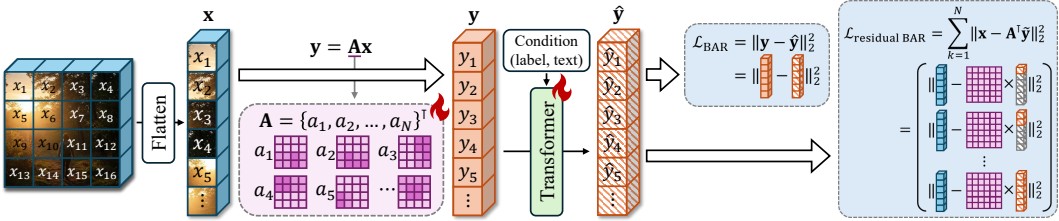

Figure 2: Pipeline of BAR and our learnable approach of transform matrix $\mathbf{A}$s. The transform matrix $\mathbf{A}$ as a learnable parameter is end-to-end optimized along with AR transformer. While $\mathcal{L}_{\text{BAR}}$ effectively trains BAR models, we further apply $\mathcal{L}_{\text{residual BAR}}$ to encourage ordered $a_k$.

Therefore, we propose the **next-basis prediction** paradigm and **Basis Autoregressive (BAR)** models with a linear transform operation that multiplies the image vector $\mathbf{x}$ by a full-rank matrix $\mathbf{A}$ and transforms it to another space $\mathcal{S}' := \mathbb{R}^{N'}$

$$\mathbf{y} := \mathbf{A}\mathbf{x}, \tag{4}$$

where $\mathbf{y} = \{y_1, y_2, \ldots, y_{N'}\}^\top \in \mathcal{S}'$ and $\mathbf{A} = \{a_1, a_2, \ldots, a_{N'}\}^\top \in \mathbb{R}^{N' \times N}$. The row vectors of $\mathbf{A}$, *e.g.* $a_k \in \mathbb{R}^N$, form the basis set of $\mathcal{S}'$. Consequently, BAR projects $\mathbf{y}$ onto the sub-spaces

$$\{\mathcal{S}'_k | \mathcal{S}'_k := \text{span}(a_k), 1 \le k \le N'\}, \tag{5}$$

and progressively predicts $y_k$, *e.g.* the projections onto each sub-space $\mathcal{S}'_k$. We apply standard AR models on new sequence $\mathbf{y}$, following with a reverse transform $\mathbf{x} = \mathbf{A}^{-1}\mathbf{y}$ back to the original space $\mathcal{S}$. BAR refactors the standard basis and sub-spaces $\{\mathcal{S}_k\}$ of vanilla AR into the transformed basis and sub-spaces $\{\mathcal{S}'_k\}$. This linear transform serves as a unified framework that allows to incorporate and generalize previous methods as different designs of the transform matrix $\mathbf{A}$.

## 3.2 SPECIAL CASES

While previous methods also propose to transform the image token sequences, they predominantly rely on human inductive bias and lack rigorous mathematical formulations. xAR introduces a *next-X prediction* paradigm where X can be instantiated as token, subsample, and scale. However, the concept remains a textual description and xAR empirically adopts cell, *e.g.* a group of adjacent tokens, as the basic unit. Here, we systematically discuss prior works as special cases of matrix $\mathbf{A}$.

**VAR** (Tian et al., 2024) leverages a coarse-to-fine strategy of *next-scale prediction*, which can be viewed as a multi-scale transform $\mathbf{A}$ with its basis $a_k$ as the average pooling of different resolutions. The sequence $\mathbf{y}$ thereby reflects the progression from global features to finer local details.

**xAR** (Ren et al., 2025) groups a grid of spatially adjacent tokens as cells and sequentially predicts them. In this case, $\mathbf{A}$ also represents the re-ordering and re-grouping of the standard basis set $\{e_k\}$, where $e_k$ and $e_l$ are grouped together if $x_k$ and $x_l$ are spatially adjacent on the 2D feature grid.

**RAR** (Yu et al., 2024b) randomly permutes the token sequence during training, and gradually anneal to normal. The matrix $\mathbf{A}$ is thereby a random permutation matrix $\mathbf{P}_\pi$ and gradually anneals to $\mathbf{I}$.

**PAR** (Wang et al., 2024) accelerates inference by predicting groups of tokens with weak dependencies, *e.g.* sub-sample, in parallel. Here, the transform matrix $\mathbf{A}$ takes a specific form of selection where $e_k$ with its corresponding $x_k$ in the same sub-sample placed together.

**FAR** (Yu et al., 2025) performs AR in the frequency domain, instead of the spatial domain, from low-frequency to high-frequency components. Similarly, the transform matrix $\mathbf{A}$ can be constructed as multi-frequency filters, where each $a_k$ is a low-pass filter with different cut-off frequencies.

**TiTok** (Yu et al., 2024c) is designed to tokenize an image into an ultra-compact 1D sequence of $M \ll N$ latent representations. The corresponding transform matrix $\mathbf{A} \in \mathbb{R}^{M \times N}$ serves as an abstraction from the long sequence $\mathbf{x} \in \mathbb{R}^{N \times d}$ into the short sequence $\mathbf{y} \in \mathbb{R}^{M \times d}$.

**FractalGen** (Li et al., 2025) proposes the AR modeling with recursive, self-similar architectures inspired by fractals. The corresponding matrix $\mathbf{A}$ would be similar to VAR with its basis $a_k$ representing hierarchical structures, generating the sequence $y_k$ at various levels of the fractal recursion.

### 3.3 LEARNABLE APPROACH

Prior AR models, as discussed above, often depend on static hand-crafted designs of specific prediction units or orders derived from inductive biases, and might lead to divergent conclusions based on similar empirical observations. Therefore, we propose an adaptive algorithm to parameterize and learn the transform matrix $\mathbf{A}$ in an end-to-end manner alongside the AR model itself.

We narrow the search space for the transform matrix $\mathbf{A}$ to enhance efficiency without loss of generality. Firstly, we follow prior methods to omit the channel dimension and treat each token as a whole, as AR typically progresses on the sequence dimension. Secondly, we consider $\mathbf{A} \in \mathbb{R}^{N \times N}$ to be square matrix, as it does not change sequence lengths and remains a minimum modification of existing AR models. The matrix $\mathbf{A}$ essentially exchanges the standard basis $\{e_k\}$ for learned basis $\{a_k\}$. Furthermore, we focus on the orthogonal matrices, a noteworthy category for the transform matrix $\mathbf{A}$, in this work. Such matrices possess desirable properties for our learnable approach, for example, they preserve the Euclidean norm of vectors, *e.g.* $\|\mathbf{y}\|_2 \equiv \|\mathbf{x}\|_2$ where $\mathbf{y} = \mathbf{A}\mathbf{x}$.

To facilitate the adaptive search for the transform matrix $\mathbf{A}$, we treat it as a learnable parameter along with AR models and derive a training objective equivalent to previous AR methods. Here we mainly focus on continuous AR models. We have the following proposition.

**Proposition 1.** *Optimizing BAR on the transformed image* $\mathbf{y}$ *and token sequence* $\{y_k\}$ *is equivalent to MAR on the original image* $\mathbf{x}$ *and token sequence* $\{x_k\}$, *i.e.* $\mathcal{L}_{\mathrm{MAR}}(z_k, y_k) = \mathcal{L}_{\mathrm{MAR}}(z_k, x_k)$.

*Proof.* Consider a reference model optimized with $\mathcal{L}_{\mathrm{MAR}}$, we rewrite eq. (1) as

$$
\begin{aligned}
\mathcal{L}_{\mathrm{MAR}}^{\mathrm{ref}}(\mathbf{x}) &= \sum_{k=1}^{N} \left\| \epsilon_k - \epsilon_\eta(x_k^t | t, z_\theta(x_{<k})) \right\|_2^2 \\
&= \sum_{k=1}^{N} \left\| \frac{\sqrt{\bar{\alpha}_t}}{\sqrt{1 - \bar{\alpha}_t}} \left( \frac{(x_k^t - \sqrt{1 - \bar{\alpha}_t}\epsilon_k)}{\sqrt{\bar{\alpha}_t}} - \frac{(x_k^t - \sqrt{1 - \bar{\alpha}_t}\epsilon_\eta)}{\sqrt{\bar{\alpha}_t}} \right) \right\|_2^2 \\
&= \sum_{k=1}^{N} \left\| \frac{\sqrt{\bar{\alpha}_t}}{\sqrt{1 - \bar{\alpha}_t}} (x_k - \hat{x}_k) \right\|_2^2 = \frac{\bar{\alpha}_t}{1 - \bar{\alpha}_t} \|(\mathbf{x} - \hat{\mathbf{x}})\|_2^2,
\end{aligned}
\tag{6}
$$

where $\hat{x}_k := \frac{1}{\sqrt{\bar{\alpha}_t}} \left( x_k^t - \sqrt{1 - \bar{\alpha}_t}\epsilon_\eta \right)$ is obtained via the reparameterization of the noise prediction $\epsilon_\eta$, and $\hat{\mathbf{x}}$ is the predicted image vector. On the other hand, we also apply MAR to our transformed sequence $\mathbf{y} = \mathbf{A}\mathbf{x}$. The correspondents of noisy tokens $x_k^t$ in the transformed space $\mathcal{S}'$ are

$$
\begin{aligned}
y_k^t = a_k^\top \mathbf{x} &= \sum_{l=1}^{N} a_{k,l} x_l^t = \sum_{l=1}^{N} a_{k,l} \left( \sqrt{\bar{\alpha}_t} x_l + \sqrt{1 - \bar{\alpha}_t}\epsilon_l \right) \\
&= \sqrt{\bar{\alpha}_t} \sum_{l=1}^{N} a_{k,l} x_l + \sqrt{1 - \bar{\alpha}_t} \sum_{l=1}^{N} a_{k,l}\epsilon_l = \sqrt{\bar{\alpha}_t} y_k + \sqrt{1 - \bar{\alpha}_t}\epsilon_k',
\end{aligned}
\tag{7}
$$

where $\epsilon_k' := \sum_{l=1}^{N} a_{k,l}\epsilon_l$ is the transformed noise. Note that if we denote $\varepsilon := \{\epsilon_1, \epsilon_2, \ldots, \epsilon_N\}$ and $\varepsilon' := \{\epsilon_1', \epsilon_2', \ldots, \epsilon_N'\}$, we have $\varepsilon' = A\varepsilon$, $\mathbb{E}(\varepsilon') = A\mathbb{E}(\varepsilon) = \mathbf{0}$, and $\Sigma_{\varepsilon'} = \mathbb{E}[(\varepsilon' - \mathbb{E}(\varepsilon'))(\varepsilon' - \mathbb{E}(\varepsilon'))^\top] = \mathbb{E}[(A\varepsilon)(A\varepsilon)^\top] = \mathbf{I}$, which suggests that $\varepsilon'$ is also composed of i.i.d. Gaussian noise. The subsequent BAR objective on the transformed image $\mathbf{y}$ is

$$
\begin{aligned}
\mathcal{L}_{\mathrm{BAR}}(\mathbf{y}) &= \sum_{k=1}^{N} \left\| \epsilon_k' - \epsilon_\eta'(y_k^t | t_k, z_\theta(y_{<k})) \right\|_2^2 \\
&= \sum_{k=1}^{N} \left\| \frac{\sqrt{\bar{\alpha}_t}}{\sqrt{1 - \bar{\alpha}_t}} (y_k - \hat{y}_k) \right\|_2^2 = \frac{\bar{\alpha}_t}{1 - \bar{\alpha}_t} \|(\mathbf{y} - \hat{\mathbf{y}})\|_2^2 \\
&= \frac{\bar{\alpha}_t}{1 - \bar{\alpha}_t} \|\mathbf{A}(\mathbf{x} - \hat{\mathbf{x}})\|_2^2 = \mathcal{L}_{\mathrm{MAR}}^{\mathrm{ref}},
\end{aligned}
\tag{8}
$$

which indicates that optimizing BAR on $\mathbf{y}$ is equivalent to optimizing the underlying MAR on $\mathbf{x}$. □

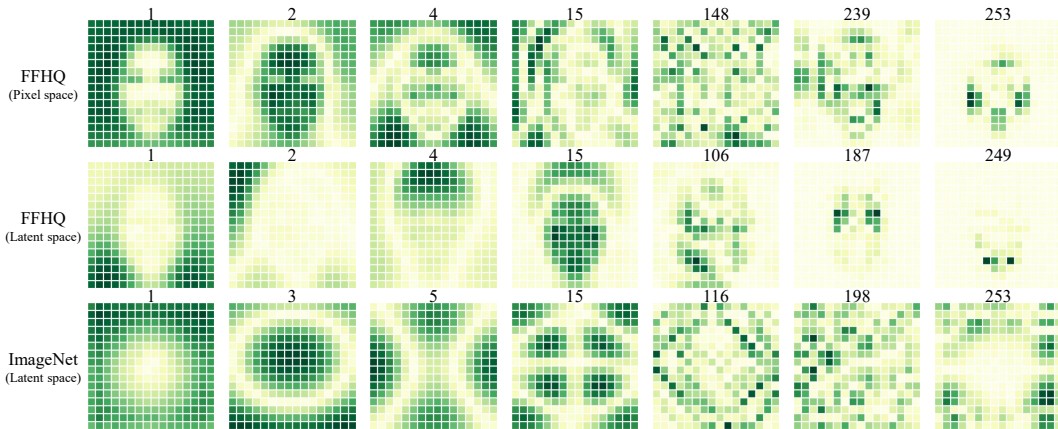

Figure 3: Visualization of learned basis $a_k$ with $k$ denoted above. Darker marks higher values.

**Proposition 2.** *Optimizing BAR on the transformed image* $\mathbf{y}$ *and token sequence* $\{y_k\}$ *is equivalent to xAR on the original image* $\mathbf{x}$ *and token sequence* $\{x_k\}$, *i.e.* $\mathcal{L}_{\text{BAR}}(\mathbf{y}) = \mathcal{L}_{\text{xAR}}(\mathbf{x})$.

Similarly, we can also train xAR models on the transformed image $\mathbf{y}$ and token sequence $\{y_k\}$, and leave the detailed proof for the following proposition in the Appendix. The propositions 1 and 2 ensure that $\mathcal{L}_{\text{BAR}}$ is a valid objective as $\mathcal{L}_{\text{MAR}}$ and $\mathcal{L}_{\text{xAR}}$, *e.g.* the performance of BAR and MAR(xAR) would be the same when optimizing only the network parameters. However, when incorporating the learnable matrix $\mathbf{A}$, we will see great improvements in the following experiments.

### 3.4 RESIDUAL OBJECTIVE

While propositions 1 and 2 indicate the potential for direct optimization of the transform matrix $\mathbf{A}$, the full exploitation of the capabilities of AR models necessitates the incorporation of certain desirable properties. Specifically, a primary objective is the maximization of information content within the earlier basis vectors $a_k$. This aims to facilitate the most accurate possible reconstruction of the image $\mathbf{x}$ utilizing earlier tokens $y_k$, a characteristic intrinsic to the sequential token prediction nature of AR models. Consequently, we introduce a residual training objective designed to explicitly enforce an ordering of the basis vectors $a_k$ and tokens $y_k$ according to their respective information richness and their contribution to the image recovery process.

Note that the BAR objective in eq. (8) can be rewritten as

$$\mathcal{L}_{\text{BAR}}(\mathbf{y}) = \frac{\bar{\alpha}_t}{1 - \bar{\alpha}_t} \left\| \mathbf{y} - \hat{\mathbf{y}} \right\|_2^2 = \frac{\bar{\alpha}_t}{1 - \bar{\alpha}_t} \left\| \mathbf{A}(\mathbf{x} - \mathbf{A}^\top \hat{\mathbf{y}}) \right\|_2^2 = \frac{\bar{\alpha}_t}{1 - \bar{\alpha}_t} \left\| \mathbf{x} - \mathbf{A}^\top \hat{\mathbf{y}} \right\|_2^2, \quad (9)$$

we propose the residual BAR objective as

$$\mathcal{L}_{\text{residual BAR}}(\mathbf{y}) = \frac{\bar{\alpha}_t}{1 - \bar{\alpha}_t} \sum_{k=1}^N \left\| \mathbf{x} - \mathbf{A}^\top \tilde{\mathbf{y}}_{\mathbf{k}} \right\|_2^2 \quad \text{where} \quad \tilde{\mathbf{y}}_{\mathbf{k}} := \hat{\mathbf{y}}^\top (\sum_{l=1}^k e_l). \quad (10)$$

Here, $\tilde{\mathbf{y}}_{\mathbf{k}}$ is the $k$-prefix of $\hat{\mathbf{y}}$, *i.e.* $\tilde{\mathbf{y}}_{\mathbf{k}} := \{\tilde{y}_1, \tilde{y}_2, \ldots, \tilde{y}_N\}$ where $\tilde{y}_i = \hat{y}_i$ for $1 \leq i \leq k$ and $\tilde{y}_i = 0$ for $k < i \leq N$. The motivation lies that the first output token $y_1$ should maximize the recovery of image $\mathbf{x}$, and the following tokens $y_k$ should maximize the recovery of the residuals $\mathbf{x} - \mathbf{A}^\top \tilde{\mathbf{y}}_{\mathbf{k-1}}$.

**Discussion** $\mathcal{L}_{\text{residual BAR}}$ partially shares the same principle as VAR and RQ-VAE. While VAR assumes the average of patches as coarse contexts and RQ-VAE progressively quantizes the remaining residuals, our design enables an adaptive learning process and introduces fewer inductive biases.

### 3.5 IMPLEMENTATION DETAILS

**Regularization** is critical to our learnable algorithm, since we assume $\mathbf{A}$ to be orthogonal. Specifically, we use the term $\mathcal{L}_{\text{reg}} := \|\mathbf{A}^\top \mathbf{A} - \mathbf{I}\|_2^2$, and train models with $\mathcal{L}_{\text{BAR}} + \mathcal{L}_{\text{reg}}$ or $\mathcal{L}_{\text{residual BAR}} + \mathcal{L}_{\text{reg}}$.

**Orthogonal projection** as closed-form solution of Orthogonal Procrustes problem (Schönemann, 1966; Ge et al., 2013) applies Singular Value Decomposition (SVD) as $\mathbf{U}\mathbf{S}\mathbf{V}^\top = \mathbf{A}$, then lets

Table 1: Benchmarking conditional image generation on ImageNet $256 \times 256$.

| Type | Model | FID↓ | IS↑ | Pre.↑ | Rec.↑ | Time↓ | #Step↓ | #Param↓ |
|------|-------|------|-----|-------|-------|-------|--------|---------|
| Diff. | ADM (Dhariwal & Nichol, 2021) | 4.59 | 186.7 | 0.82 | 0.52 | 44.68 | 250 | 544M |
| Diff. | LDM (Rombach et al., 2022) | 3.60 | 247.7 | 0.87 | 0.48 | 207.2 | 250 | 400M |
| Diff. | U-ViT (Bao et al., 2023) | 2.29 | 263.9 | 0.82 | 0.57 | - | - | 501M |
| Diff. | DiT (Peebles & Xie, 2023) | 2.27 | 278.2 | 0.83 | 0.57 | 11.97 | 250 | 675M |
| Diff. | SiT (Ma et al., 2024) | 2.06 | 277.5 | 0.83 | 0.59 | 11.97 | 250 | 675M |
| Diff. | VDM++ (Kingma & Gao, 2023) | 2.12 | 267.7 | 0.81 | 0.65 | - | - | 2.0B |
| Diff. | MDTv2 (Gao et al., 2023) | 1.58 | 314.7 | 0.79 | 0.65 | - | 250 | 676M |
| Diff. | REPA (Yu et al., 2024d) | 1.42 | 305.7 | 0.80 | 0.65 | 11.97 | 250 | 675M |
| Diff. | Light.DiT (Yao & Wang, 2025) | 1.35 | 295.3 | 0.79 | 0.65 | - | 250 | 675M |
| Diff. | MG (Tang et al., 2025) | 1.34 | 321.5 | 0.81 | 0.65 | 6.03 | 250 | 675M |
| Mask. | MaskGIT (Chang et al., 2022) | 6.18 | 182.1 | 0.80 | 0.51 | 0.5 | 8 | 227M |
| Mask. | RCG (Li et al., 2024a) | 3.49 | 215.5 | - | - | 1.9 | 20 | 502M |
| Mask. | TiTok (Yu et al., 2024c) | 1.97 | 281.8 | - | - | - | 64 | 287M |
| AR | VQGAN (Esser et al., 2021) | 5.20 | 280.3 | - | - | 6.38 | 256 | 1.4B |
| AR | RQTrans. (Lee et al., 2022) | 3.80 | 323.7 | - | - | 5.58 | 256 | 3.8B |
| AR | FAR (Yu et al., 2025) | 3.21 | 300.6 | 0.81 | 0.55 | - | 10 | 812M |
| AR | PAR (Wang et al., 2024) | 2.29 | 255.5 | 0.82 | 0.58 | 3.46 | 147 | 3.1B |
| AR | LlamaGen (Sun et al., 2024) | 2.18 | 263.3 | 0.81 | 0.58 | 12.41 | 576 | 3.1B |
| AR | VAR (Tian et al., 2024) | 1.73 | 350.2 | 0.82 | 0.60 | 0.27 | 10 | 2.0B |
| AR | FlowAR (Ren et al., 2024) | 1.65 | 296.5 | 0.83 | 0.60 | - | 10 | 1.9B |
| AR | MAR (Li et al., 2024b) | 1.55 | 303.7 | 0.81 | 0.62 | 28.24 | 64 | 943M |
| AR | RAR (Yu et al., 2024b) | 1.48 | 326.0 | 0.80 | 0.63 | - | 256 | 1.5B |
| AR | xAR (Ren et al., 2025) | 1.24 | 301.6 | 0.83 | 0.64 | 0.68 | 50 | 1.1B |
| AR | BAR-B$_{(ours)}$ | 1.56 | 292.4 | 0.83 | 0.63 | 0.08 | 50 | 172M |
| AR | BAR-L$_{(ours)}$ | 1.21 | 301.1 | 0.84 | 0.64 | 0.27 | 50 | 608M |
| AR | BAR-H$_{(ours)}$ | **1.15** | 327.1 | 0.86 | 0.68 | 0.68 | 50 | 1.1B |

$\mathbf{A} = \mathbf{U}\mathbf{S}_\delta\mathbf{V}^\top$, where the diagonals of $\mathbf{S}$ is clamped to $(1 - \delta, 1 + \delta)$ as $\mathbf{S}_\delta$ with $\delta = 0$ in hard projection and $\delta \in (0, 1)$ in soft projection.

**Initialization** of the transform matrix $\mathbf{A}$ is important for its optimization. While the identity matrix $\mathbf{I}$ corresponds to vanilla AR, we can also use a random matrix followed by an orthogonal projection.

## 4 EXPERIMENT

### 4.1 SETUP

**Implementation.** We apply our BAR framework to two different architectures of AR and follow the protocol in MAR (Li et al., 2024b) and xAR (Ren et al., 2025). We primarily conduct experiments on the ImageNet (Deng et al., 2009) $256 \times 256$ dataset and then raise the spatial resolution of images to $512 \times 512$. We also conduct ablation studies on the FFHQ (Karras et al., 2019) dataset for its human-face-centric nature at $64 \times 64$ and $256 \times 256$ resolutions. The KL-16 continuous tokenizer provided by LDM (Rombach et al., 2022) is employed to encode the images into latent tokens. Our models are mainly based on xAR (Ren et al., 2025), and we conduct ablation experiments with xAR-B. All models are trained following previous settings (Li et al., 2024b; Ren et al., 2025), *e.g.* 800 epochs and the batch size of 256, for fair comparisons, and all experiments are conducted with 16 NVIDIA A100 40G GPUs. For the text-to-image generation task, we follow FAR (Yu et al., 2025) and use the JourneyDB (Sun et al., 2023) dataset and the Qwen2-1.5B (Yang et al., 2024) text encoder.

**Evaluation.** As metrics, we report the commonly adopted Frechet inception distance (FID) (Heusel et al., 2017), Inception Score (IS) (Salimans et al., 2016), Precision (Pre.), and Recall (Rec.) (Kynkäänniemi et al., 2019) over 50K generated samples, which is consistent with previous works. We also list the number of parameters, sampling steps, and the wall-clock time to generate one image for each method, which facilitate the thorough investigation of the balance between sampling quality and generation efficiency. For the text-to-image generation task, we follow FAR (Yu et al., 2025) to measure FID-30K on MS-COCO 2014 (Lin et al., 2014) and the GenEval (Ghosh et al., 2023) benchmark.

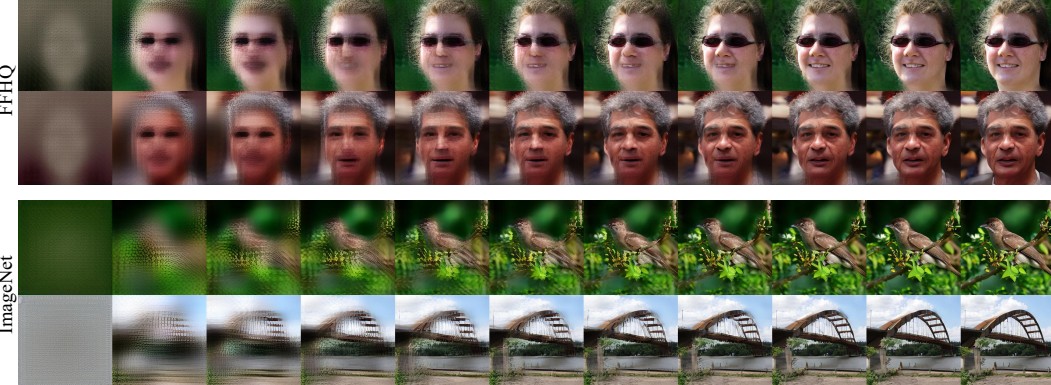

Figure 4: We decode the generated sequences every 25 tokens. The first column uses only one token.

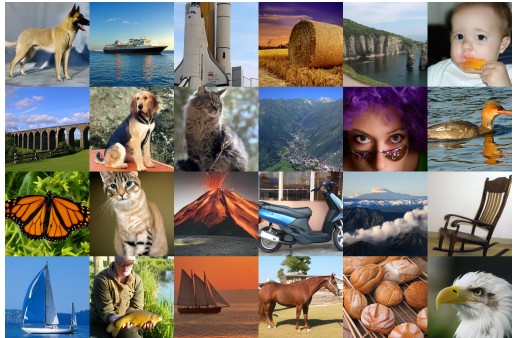

Figure 5: Uncurated samples on ImageNet 256.

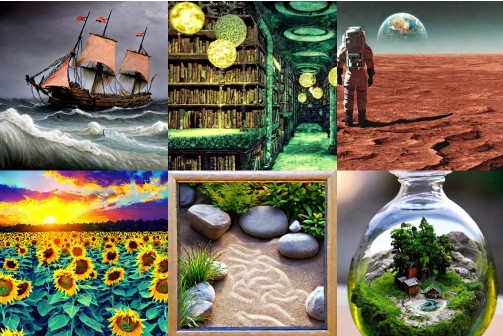

Figure 6: Generated text-to-image samples.

## 4.2 MAIN RESULTS

**ImageNet** $256 \times 256$ **benchmark**. In table 1, we present the system-level comparison of conditional generation performance on the ImageNet $256 \times 256$ dataset with concurrent advanced methods, including diffusion, mask-based, and AR models. Notably, we achieve new state-of-the-art results with an FID of 1.15, while our BAR-B model also exhibits strong performance with only 172M parameters and a lightning speed of 0.08 seconds per image. While our models offer outstanding generation quality, they are also faster than most of the previous methods. The generated images of our model are provided in fig. 5.

**Compatibility with different architectures**. While our models are mainly based on xAR, we show in table 2 that the framework of BAR is also compatible with other existing AR models. We additionally apply our method to MAR and experiment with their -B, -L, and -H variants. The significant improvement over their corresponding baselines indicates the effectiveness of BAR on different AR approaches and model sizes.

**Scalability to ImageNet** $512 \times 512$. The extension of our method to high-resolution images is also notable. We conduct experiments on the ImageNet $512 \times 512$ dataset in table 3 and the remarkable margin over both MAR and xAR baselines demonstrates that BAR is capable of modeling challenging distributions and longer token sequences.

**Text-to-image generation**. In addition to the class-conditioned task on ImageNet, we also employ BAR for text-to-image generation and depict the comparison in table 4. Following FAR (Yu et al., 2025), our model further surpasses it by a gain of 1.36 FID. It suggests that BAR is promising across various conditioning modalities. We also present the generated samples of our method in fig. 6.

## 4.3 ABLATION STUDY

**Visualizing learned basis**. We visualize the learned basis $a_k$ in fig. 3, where $k$ is denoted above each subfigure. The basis $a_k$ is reshaped to match the 2D feature grid of images, and darker regions mark high values, *i.e.* the weight of corresponding image patch is higher for $a_k$. Here, we additionally experiment on pixel-space $64 \times 64$ FFHQ and patchify $4 \times 4$ pixels as a token. The basis on pixel-

Table 2: Experiments on different models.

| Model | FID↓ | IS↑ | Pre.↑ | Rec.↑ |
|---|---|---|---|---|
| MAR-B (Li et al., 2024b) | 2.31 | 281.7 | 0.82 | 0.57 |
| +BAR(ours) | 2.18 | 289.6 | 0.83 | 0.60 |
| MAR-L (Li et al., 2024b) | 1.78 | 296.0 | 0.81 | 0.60 |
| +BAR(ours) | 1.56 | 304.1 | 0.83 | 0.61 |
| MAR-H (Li et al., 2024b) | 1.55 | 303.7 | 0.81 | 0.62 |
| +BAR(ours) | 1.49 | 312.8 | 0.82 | 0.64 |
| xAR-B (Ren et al., 2025) | 1.72 | 280.4 | 0.82 | 0.59 |
| +BAR(ours) | 1.63 | 291.3 | 0.83 | 0.62 |
| xAR-L (Ren et al., 2025) | 1.28 | 292.5 | 0.82 | 0.62 |
| +BAR(ours) | 1.24 | 299.7 | 0.84 | 0.64 |
| xAR-H (Ren et al., 2025) | 1.24 | 301.6 | 0.83 | 0.64 |
| +BAR(ours) | 1.15 | 324.5 | 0.85 | 0.68 |

Table 3: Experiments on ImageNet 512.

| Model | FID↓ | IS↑ | Pre.↑ | Rec.↑ |
|---|---|---|---|---|
| VAR (Tian et al., 2024) | 2.63 | 303.2 | 0.82 | 0.62 |
| REPA (Yu et al., 2024d) | 2.08 | 274.6 | 0.81 | 0.61 |
| EDM2 (Karras et al., 2024) | 1.81 | 273.2 | 0.85 | 0.63 |
| MAR-L (Li et al., 2024b) | 1.73 | 279.9 | 0.84 | 0.62 |
| +BAR(ours) | 1.65 | 287.7 | 0.85 | 0.64 |
| xAR-L (Ren et al., 2025) | 1.70 | 281.5 | 0.84 | 0.64 |
| +BAR(ours) | 1.63 | 292.0 | 0.85 | 0.64 |

Table 4: Experiments on text-to-image.

| Model | FID↓ | GenEval↑ |
|---|---|---|
| LDM (Rombach et al., 2022) | 12.70 | 0.37 |
| LlamaGen (Sun et al., 2024) | 15.05 | 0.32 |
| FAR (Yu et al., 2025) | 13.91 | 0.37 |
| BAR | 12.55 | 0.39 |

Table 5: Ablation on initialization.

| Initialize | FID↓ | IS↑ | Pre.↑ | Rec.↑ |
|---|---|---|---|---|
| Baseline | 1.72 | 280.4 | 0.82 | 0.59 |
| Identity | 1.63 | 291.3 | 0.83 | 0.62 |
| Orthogonal | 1.66 | 289.6 | 0.83 | 0.61 |

Table 6: Ablation on orthogonal projection.

| Projection | FID↓ | IS↑ | Pre.↑ | Rec.↑ |
|---|---|---|---|---|
| Baseline | 1.72 | 280.4 | 0.82 | 0.59 |
| None | 1.70 | 284.9 | 0.81 | 0.61 |
| Hard | 1.66 | 285.8 | 0.82 | 0.61 |
| $\text{Soft}_{\delta=0.5}$ | 1.63 | 291.3 | 0.83 | 0.62 |

Table 7: Ablation on training objective.

| Objective | FID↓ | IS↑ | Pre.↑ | Rec.↑ |
|---|---|---|---|---|
| Baseline | 1.72 | 280.4 | 0.82 | 0.59 |
| $\mathcal{L}_{\text{BAR}}$ | 1.64 | 289.7 | 0.84 | 0.64 |
| $\mathcal{L}_{\text{residual BAR}}$ | 1.63 | 291.3 | 0.83 | 0.62 |

space FFHQ clearly reflects the shape of human faces, while the basis on latent-space FFHQ exhibits less continuity, which explains the success of AR models on tokenized images. Furthermore, the earlier basis on ImageNet shows an interesting pattern, while later ones seem to be more random, which is beyond static hand-crafted designs.

**Visualizing generation process**. We also visualize the generation process of BAR models by progressively decoding the generated token sequence with its prefix, *e.g.* we only decode the first $k$ tokens with $k$ increasing with a step size of 25 starting from the first token. The output images also show a coarse-to-fine paradigm, which is consistent with the motivation of our residual objective.

**Initialization strategy**. The initialization of transform matrix $\mathbf{A}$ is critical to our learnable approach. In table 5, using the identity matrix $\mathbf{I}$ as initialization obtains using best results, since it corresponds to vanilla AR models, while our method also offers a gain over the baseline with random orthogonal initialization.

**Orthogonal projection**. The orthogonal projection is also crucial to the training process. In table 6, the soft projection with $\delta = 0.5$ offers the best results, while the performance is significantly impaired without projection. The main reason is that the regularization term for orthogonality alone is not strong enough, while hard projection limits the potential update directions for $\mathbf{A}$.

**Training objective**. Although $\mathcal{L}_{\text{BAR}}$ itself is capable of effectively training BAR models, we further introduce and apply $\mathcal{L}_{\text{residual BAR}}$ that encourages $a_k$ to be ordered by the recovery of the original images and mimics a coarse-to-fine characteristic as in fig. 4. The results in table 7 confirm that both $\mathcal{L}_{\text{BAR}}$ and $\mathcal{L}_{\text{residual BAR}}$ provide satisfactory performance, while $\mathcal{L}_{\text{residual BAR}}$ is slightly better.

## 5  CONCLUSION

In conclusion, this work introduces Basis Autoregressive (BAR), a novel paradigm for image generation that addresses the inherent limitations of traditional AR models tied to fixed, raster-scan prediction of tokens. By conceptualizing tokens as the projection of image vector on basis of linear space and employing an end-to-end learnable transform of these bases, BAR offers a unified mathematical framework. It not only encompasses previous methodologies as specific instances but also adaptively optimizes the transform to discover effective strategies beyond manual designs. The demonstrated state-of-the-art performance, highlighted by an FID score of 1.15 on the ImageNet-256 benchmark, underscores its capability to transcend human biases and significantly advance the field of image generation, including its application in text-to-image synthesis. BAR represents a significant step in developing more flexible and powerful AR models for visual content creation.

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

# A  PROOFS

## A.1  MAR VARIANTS

In this section, we provide further details of propositions 1 and 2 and their corresponding proofs. We first begin with BAR implemented with existing MAR architectures.

**Proposition 1.** *Optimizing BAR on the transformed image* $\mathbf{y}$ *and token sequence* $\{y_k\}$ *is equivalent to MAR on the original image* $\mathbf{x}$ *and token sequence* $\{x_k\}$, *i.e.* $\mathcal{L}_{\text{MAR}}(z_k, y_k) = \mathcal{L}_{\text{MAR}}(z_k, x_k)$.

*Proof.* Consider a reference model optimized with $\mathcal{L}_{\text{MAR}}$. Recall that the noise-corrupted sample is obtained via the interpolating ground-truth image and sampled noise via the noise schedule

$$x_k^t = \sqrt{\bar{\alpha}_t} x_k + \sqrt{1 - \bar{\alpha}_t} \epsilon, \tag{11}$$

where $\epsilon \sim \mathcal{N}(0, 1)$ is random noise drawn from Gaussian distribution, and $\bar{\alpha}$ is the noise schedule used in diffusion models (Song & Ermon, 2019; Ho et al., 2020). Given the model outputs as predictions of added noises, we can obtain the prediction of unnoised image $\hat{\mathbf{x}}$ via the reparameterization

$$\hat{x}_k = \frac{x_k^t - \sqrt{1 - \bar{\alpha}_t} \hat{\epsilon}}{\sqrt{\bar{\alpha}_t}}, \quad \text{where} \quad \hat{\epsilon} := \epsilon_\eta(x_k^t | t, z_\theta(x_{<k})). \tag{12}$$

We thereby rewrite eq. (1) as

$$
\begin{aligned}
\mathcal{L}_{\text{MAR}}^{\text{ref}}(\mathbf{x}) &= \sum_{k=1}^{N} \left\| \epsilon_k - \epsilon_\eta(x_k^t | t, z_\theta(x_{<k})) \right\|_2^2 \\
&= \sum_{k=1}^{N} \left\| \frac{1}{\sqrt{1 - \bar{\alpha}_t}} \left( \sqrt{1 - \bar{\alpha}_t} \epsilon_k - \sqrt{1 - \bar{\alpha}_t} \epsilon_\eta \right) \right\|_2^2 \\
&= \sum_{k=1}^{N} \left\| \frac{1}{\sqrt{1 - \bar{\alpha}_t}} \left( (x_k^t - \sqrt{1 - \bar{\alpha}_t} \epsilon_k) - (x_k^t - \sqrt{1 - \bar{\alpha}_t} \epsilon_\eta) \right) \right\|_2^2 \\
&= \sum_{k=1}^{N} \left\| \frac{\sqrt{\bar{\alpha}_t}}{\sqrt{1 - \bar{\alpha}_t}} \left( \frac{(x_k^t - \sqrt{1 - \bar{\alpha}_t} \epsilon_k)}{\sqrt{\bar{\alpha}_t}} - \frac{(x_k^t - \sqrt{1 - \bar{\alpha}_t} \epsilon_\eta)}{\sqrt{\bar{\alpha}_t}} \right) \right\|_2^2 \\
&= \sum_{k=1}^{N} \left\| \frac{\sqrt{\bar{\alpha}_t}}{\sqrt{1 - \bar{\alpha}_t}} (x_k - \hat{x}_k) \right\|_2^2 \\
&= \frac{\bar{\alpha}_t}{1 - \bar{\alpha}_t} \left\| (\mathbf{x} - \hat{\mathbf{x}}) \right\|_2^2.
\end{aligned}
\tag{13}
$$

On the other hand, we also apply MAR to our transformed sequence $\mathbf{y} = \mathbf{A}\mathbf{x}$. The correspondents of noisy tokens $x_k^t$ in the transformed space $\mathcal{S}'$ are

$$
\begin{aligned}
y_k^t &= a_k^\top \mathbf{x} \\
&= \sum_{l=1}^{N} a_{k,l} x_l^t \\
&= \sum_{l=1}^{N} a_{k,l} \left( \sqrt{\bar{\alpha}_t} x_l + \sqrt{1 - \bar{\alpha}_t} \epsilon_l \right) \\
&= \sqrt{\bar{\alpha}_t} \sum_{l=1}^{N} a_{k,l} x_l + \sqrt{1 - \bar{\alpha}_t} \sum_{l=1}^{N} a_{k,l} \epsilon_l \\
&= \sqrt{\bar{\alpha}_t} y_k + \sqrt{1 - \bar{\alpha}_t} \epsilon_k',
\end{aligned}
\tag{14}
$$

where the noise added to transformed tokens $y_k$ is

$$\epsilon'_k := \sum_{l=1}^{N} a_{k,l}\epsilon_l. \tag{15}$$

Note that if we concatenate all noises into a single vector and denote $\varepsilon := \{\epsilon_1, \epsilon_2, \ldots, \epsilon_N\}$ and $\varepsilon' := \{\epsilon'_1, \epsilon'_2, \ldots, \epsilon'_N\}$, we find that first sampling $\varepsilon \sim \mathcal{N}(0, 1)$ then using the transform in eq. (15), *i.e.* $\varepsilon' = \mathbf{A}\varepsilon$, to obtain $\varepsilon'$ is equivalent to directly sampling from $\varepsilon' \sim \mathcal{N}(0, 1)$. To validate this, we can calculate the expectation of $\varepsilon'$

$$\mathbb{E}(\varepsilon') = \mathbb{E}(\mathbf{A}\varepsilon) = \mathbf{A}\mathbb{E}(\varepsilon) = \mathbf{0}, \tag{16}$$

and its covariance

$$\begin{aligned}
\Sigma_{\varepsilon'} &= \mathbb{E}[(\varepsilon' - \mathbb{E}(\varepsilon'))(\varepsilon' - \mathbb{E}(\varepsilon'))^\top] \\
&= \mathbb{E}[\varepsilon'\varepsilon'^\top] \\
&= \mathbb{E}[(\mathbf{A}\varepsilon)(\mathbf{A}\varepsilon)^\top] \\
&= \mathbb{E}[\mathbf{A}\varepsilon\varepsilon^\top\mathbf{A}^\top] \\
&= \mathbf{A}\mathbb{E}[\varepsilon\varepsilon^\top]\mathbf{A}^\top \\
&= \mathbf{A}\mathbb{E}[(\varepsilon - \mathbb{E}(\varepsilon))(\varepsilon - \mathbb{E}(\varepsilon))^\top]\mathbf{A}^\top \\
&= \mathbf{A}\Sigma_\varepsilon\mathbf{A}^\top \\
&= \mathbf{A}\mathbf{A}^\top \\
&= \mathbf{I},
\end{aligned} \tag{17}$$

which suggest that $\varepsilon'$ is also composed of i.i.d. Gaussian noise and can be directly sampled with $\varepsilon' \sim \mathcal{N}(0, 1)$. This ensures that we can directly apply MAR framework on the transformed image $\mathbf{y}$. The subsequent BAR objective is

$$\begin{aligned}
\mathcal{L}_{\text{BAR}}(\mathbf{y}) &= \sum_{k=1}^{N} \left\| \epsilon'_k - \epsilon'_\eta(y_k^t|t_k, z_\theta(y_{<k})) \right\|_2^2 \\
&= \sum_{k=1}^{N} \left\| \frac{1}{\sqrt{1 - \bar{\alpha}_t}} \left( \sqrt{1 - \bar{\alpha}_t}\epsilon'_k - \sqrt{1 - \bar{\alpha}_t}\epsilon'_\eta \right) \right\|_2^2 \\
&= \sum_{k=1}^{N} \left\| \frac{1}{\sqrt{1 - \bar{\alpha}_t}} \left( (y_k^t - \sqrt{1 - \bar{\alpha}_t}\epsilon'_k) - (y_k^t - \sqrt{1 - \bar{\alpha}_t}\epsilon'_\eta) \right) \right\|_2^2 \\
&= \sum_{k=1}^{N} \left\| \frac{\sqrt{\bar{\alpha}_t}}{\sqrt{1 - \bar{\alpha}_t}} \left( \frac{(y_k^t - \sqrt{1 - \bar{\alpha}_t}\epsilon'_k)}{\sqrt{\bar{\alpha}_t}} - \frac{(y_k^t - \sqrt{1 - \bar{\alpha}_t}\epsilon'_\eta)}{\sqrt{\bar{\alpha}_t}} \right) \right\|_2^2 \\
&= \sum_{k=1}^{N} \left\| \frac{\sqrt{\bar{\alpha}_t}}{\sqrt{1 - \bar{\alpha}_t}} (y_k - \hat{y}_k) \right\|_2^2 \\
&= \frac{\bar{\alpha}_t}{1 - \bar{\alpha}_t} \left\| (\mathbf{y} - \hat{\mathbf{y}}) \right\|_2^2 \\
&= \frac{\bar{\alpha}_t}{1 - \bar{\alpha}_t} \left\| \mathbf{A}(\mathbf{x} - \hat{\mathbf{x}}) \right\|_2^2 \\
&= \frac{\bar{\alpha}_t}{1 - \bar{\alpha}_t} \left\| (\mathbf{x} - \hat{\mathbf{x}}) \right\|_2^2 \\
&= \mathcal{L}_{\text{MAR}}^{\text{ref}},
\end{aligned} \tag{18}$$

which indicates that optimizing BAR on $\mathbf{y}$ is equivalent to optimizing the underlying MAR on $\mathbf{x}$. $\quad\square$

## A.2 XAR VARIANTS

Then, we discuss the case where BAR is implemented with the model architectures of xAR. and provide the detailed proof of proposition 2.

**Proposition 2.** *Optimizing BAR on the transformed image* $\mathbf{y}$ *and token sequence* $\{y_k\}$ *is equivalent to xAR on the original image* $\mathbf{x}$ *and token sequence* $\{x_k\}$, *i.e.* $\mathcal{L}_{\mathrm{BAR}}(\mathbf{y}) = \mathcal{L}_{\mathrm{xAR}}(\mathbf{x})$.

*Proof.* Consider a reference model optimized with $\mathcal{L}_{\mathrm{xAR}}$. Recall that the noise-corrupted sample is obtained via the interpolating ground-truth image and sampled noise by

$$x_k^{t_k} = (1 - t_k)x_k + t_k\epsilon_k, \tag{19}$$

where $\epsilon_k \sim \mathcal{N}(0, 1)$ is Gaussian noise, and the ground-truth flow is

$$v_k^{t_k} = \frac{dx_k^{t_k}}{dt_k} = \epsilon_k - x_k. \tag{20}$$

Similarly, we can obtain the prediction of unnoised image $\hat{\mathbf{x}}$ via the reparameterization

$$\hat{x}_k = \epsilon_k - \hat{v}_k^{t_k}, \quad \text{where} \quad \hat{v}_k^{t_k} := v_\theta(\{x_1^{t_1}, x_2^{t_2}, \ldots, x_k^{t_k}\}, t_k). \tag{21}$$

We thereby rewrite eq. (2) as

$$
\begin{aligned}
\mathcal{L}_{\mathrm{xAR}}^{\mathrm{ref}} &= \sum_{k=1}^{N} \left\| v_\theta(\{x_1^{t_1}, x_2^{t_2}, \ldots, x_k^{t_k}\}, t_k) - v_k^{t_k} \right\|_2^2 \\
&= \sum_{k=1}^{N} \left\| (\epsilon_k - \hat{x}_k) - (\epsilon_k - x_k) \right\|_2^2 \\
&= \sum_{k=1}^{N} \left\| \hat{x}_k - x_k \right\|_2^2 \\
&= \left\| \hat{\mathbf{x}} - \mathbf{x} \right\|_2^2 .
\end{aligned}
\tag{22}
$$

On the other hand, we also apply xAR to our transformed sequence $\mathbf{y} = \mathbf{A}\mathbf{x}$. The correspondents of noisy tokens $x_k^t$ in the transformed space $\mathcal{S}'$ are

$$
\begin{aligned}
y_k^t &= a_k^\top \mathbf{x} \\
&= \sum_{l=1}^{N} a_{k,l} x_l^t \\
&= \sum_{l=1}^{N} a_{k,l} \left( (1 - t_l)x_l + t_l\epsilon_l \right) \\
&= (1 - t_k) \sum_{l=1}^{N} a_{k,l} x_l + t_k \sum_{l=1}^{N} a_{k,l}\epsilon_l \\
&= (1 - t_k)y_k + t_k\epsilon_k',
\end{aligned}
\tag{23}
$$

where we assume that $\forall i \in \{1, 2, \ldots, k\}$ and $j \in \{1, 2, \ldots, k\}, t_i = t_j$ for simplicity, since $\{t\} = \{t_1, t_2, \ldots, t_n\} \sim \mathbf{U}[0, 1]$ is arbitrarily sampled in xAR (Ren et al., 2025). The noise added to transformed tokens $y_k$ is

$$\epsilon_k' := \sum_{l=1}^{N} a_{k,l}\epsilon_l. \tag{24}$$

Note that if we concatenate all noises into a single vector and denote $\varepsilon := \{\epsilon_1, \epsilon_2, \ldots, \epsilon_N\}$ and $\varepsilon' := \{\epsilon_1', \epsilon_2', \ldots, \epsilon_N'\}$, we find that first sampling $\varepsilon \sim \mathcal{N}(0, 1)$ then using the transform in eq. (24), *i.e.* $\varepsilon' = \mathbf{A}\varepsilon$, to obtain $\varepsilon'$ is equivalent to directly sampling from $\varepsilon' \sim \mathcal{N}(0, 1)$. To validate this, we

can calculate the expectation of $\varepsilon'$

$$\begin{aligned}
\mathbb{E}(\varepsilon') &= \mathbb{E}(\mathbf{A}\varepsilon) \\
&= \mathbf{A}\mathbb{E}(\varepsilon) \\
&= \mathbf{0},
\end{aligned} \tag{25}$$

and its covariance

$$\begin{aligned}
\Sigma_{\varepsilon'} &= \mathbb{E}[(\varepsilon' - \mathbb{E}(\varepsilon'))(\varepsilon' - \mathbb{E}(\varepsilon'))^{\top}] \\
&= \mathbb{E}[\varepsilon'\varepsilon'^{\top}] \\
&= \mathbb{E}[(\mathbf{A}\varepsilon)(\mathbf{A}\varepsilon)^{\top}] \\
&= \mathbb{E}[\mathbf{A}\varepsilon\varepsilon^{\top}\mathbf{A}^{\top}] \\
&= \mathbf{A}\mathbb{E}[\varepsilon\varepsilon^{\top}]\mathbf{A}^{\top} \\
&= \mathbf{A}\mathbb{E}[(\varepsilon - \mathbb{E}(\varepsilon))(\varepsilon - \mathbb{E}(\varepsilon))^{\top}]\mathbf{A}^{\top} \\
&= \mathbf{A}\Sigma_{\varepsilon}\mathbf{A}^{\top} \\
&= \mathbf{A}\mathbf{A}^{\top} \\
&= \mathbf{I},
\end{aligned} \tag{26}$$

which suggest that $\varepsilon'$ is also composed of i.i.d. Gaussian noise and can be directly sampled with $\varepsilon' \sim \mathcal{N}(0, 1)$. This ensures that we can directly apply xAR framework on the transformed image $\mathbf{y}$. The subsequent BAR objective is

$$\begin{aligned}
\mathcal{L}_{\text{BAR}} &= \sum_{k=1}^{N} \left\| v'_{\theta}(\{y_1^{t_1}, y_2^{t_2}, \ldots, y_k^{t_k}\}, t_k) - v'^{t_k}_k \right\|_2^2 \\
&= \sum_{k=1}^{N} \|(\epsilon'_k - \hat{y}_k) - (\epsilon'_k - y_k)\|_2^2 \\
&= \sum_{k=1}^{N} \|\hat{y}_k - y_k\|_2^2 \\
&= \|\hat{\mathbf{y}} - \mathbf{y}\|_2^2 \\
&= \|\mathbf{A}(\hat{\mathbf{x}} - \mathbf{x})\|_2^2, \\
&= \|\hat{\mathbf{x}} - \mathbf{x}\|_2^2 \\
&= \mathcal{L}_{\text{xAR}}^{\text{ref}},
\end{aligned} \tag{27}$$

where the ground-truth flow of $y_k^{t_k}$ is

$$v'^{t_k}_k = \frac{dy_k^{t_k}}{dt_k} = \epsilon'_k - y_k. \tag{28}$$

$\square$

## B  LIMITATIONS AND BROADER IMPACTS

**Limitations** This work focuses on enhancing the output quality and training speed of current AR models, but still depends on existing VAEs (Rombach et al., 2022). Recent studies (Yao & Wang, 2025; Leng et al., 2025) achieved significant success with the end-to-end training of VAE and AR models, and incorporating our approach into VAE would be a promising direction. Furthermore, this work are mainly conducted on continuous AR models. Extending BAR to discrete AR models is an imperative next-step, as we discussed in section E.

**Broader Impacts** Although this work primarily discussed images generation AR models, the next-token prediction paradigm is employed on other modalities, *e.g.* video and speech. Applying this work to AR models in these fields shows potential. On the downside, since our models are trained on existing dataset, it might unintentionally reproduce any biases within. Furthermore, the image

generation abilities developed could potentially be misused to create and spread false information. We will consider restricting the access to our model weights to address this.

Table 8: Setup for table 1.

| | BAR-B | BAR-L | BAR-H |
|---|---|---|---|
| **Architecture** | | | |
| Tokenizer | KL-16 (Rombach et al., 2022) | KL-16 (Rombach et al., 2022) | KL-16 (Rombach et al., 2022) |
| Input dimension | $16 \times 16 \times 16$ | $16 \times 16 \times 16$ | $16 \times 16 \times 16$ |
| Encoder layers | 8 | 16 | 20 |
| Encoder dimension | 768 | 1024 | 1280 |
| Encoder heads | 12 | 16 | 16 |
| Decoder layers | 8 | 16 | 20 |
| Decoder dimension | 768 | 1024 | 1280 |
| Decoder heads | 12 | 16 | 16 |
| Number of parameters | 172M | 608M | 1.1B |
| **Hyperparameters** | | | |
| Optimizer | AdamW (Kingma, 2014; Loshchilov, 2019) | AdamW (Kingma, 2014; Loshchilov, 2019) | AdamW (Kingma, 2014; Loshchilov, 2019) |
| Momentum $(\beta_1, \beta_2)$ | (0.9, 0.96) | (0.9, 0.96) | (0.9, 0.96) |
| Weight decay | 0.02 | 0.02 | 0.02 |
| Batch size | 2048 | 2048 | 2048 |
| Learning rate schedule | cosine | cosine | cosine |
| Peak learning rate | $4 \times 10^{-4}$ | $4 \times 10^{-4}$ | $4 \times 10^{-4}$ |
| End learning rate | $1 \times 10^{-5}$ | $1 \times 10^{-5}$ | $1 \times 10^{-5}$ |
| Total epochs | 800 | 800 | 800 |
| Warmup epochs | 100 | 100 | 100 |
| Dropout rate | 0.1 | 0.1 | 0.1 |
| Label dropout rate | 0.1 | 0.1 | 0.1 |
| **Inference** | | | |
| Sampler | Euler-Maruyama | Euler-Maruyama | Euler-Maruyama |
| Steps | 50 | 50 | 50 |

**Algorithm 1** Training BAR: PyTorch-like Pseudo-code

```python
def train(residual_loss=True):
    for step, (x, c) in enumerate(dataloader):
        # sample random noise and timestep
        noise = torch.randn(x.shape)
        timestep = torch.rand(x.shape[0], 1, 1)

        # BAR: linear transform
        y = x @ self.A_mat

        # sample y_t from y
        y_t = (1 - timestep) * y + timestep * noise

        # predict v_hat and y_hat from y_t
        v_hat = net(y_t, timestep, c)
        y_hat = noise - v_hat

        # compute loss
        if residual_loss:
            x_hat = [y_hat[:, :, k+1] @ self.A_mat[:, :k+1] for k in range(x.
                shape[-1])]
            loss = ((torch.stack(x_hat, dim=1) - x.unsqueeze(1)) ** 2).mean()
        else:
            loss = ((y_hat - y) ** 2).mean()

        # regularize term
        loss += ((self.A_mat @ self.A_mat.T - torch.eye(self.A_mat.shape[0]))
            ** 2).mean()

        # optimize
        opt.step()
        opt.zero_grad()
```

## C  HYPERPARAMETER AND IMPLEMENTATION DETAILS

**Implementations.** We implement our method based on the code of xAR (Ren et al., 2025) and MAR (Li et al., 2024b). We use the exact same structure and same hyper-parameters as xAR (Ren et al., 2025) and MAR (Li et al., 2024b) throughout all experiments. We use a batch size of 2048 in consistence with xAR (Ren et al., 2025) and MAR (Li et al., 2024b), and we apologize for the typo in the main text of the batch size of 256. We use AdamW (Kingma, 2014; Loshchilov, 2019)

with cosine learning rate schedule starting from $4 \times 10^{-4}$, warm up for 100 epochs, and gradually decay to $1 \times 10^{-5}$. The optimizer momentum is $(\beta_1, \beta_2) = (0.9, 0.96)$ and the weight decay is $0.02$. We also pre-compute and save the latent vectors of images and use these latent vectors for training, similar to xAR (Ren et al., 2025) and MAR (Li et al., 2024b). Therefore, we only apply simple random horizontal flip as data augmentation. We use continuous KL-16 VAE from LDM (Rombach et al., 2022) for encoding and decoding images. The detailed hyper-parameter setup are provided in table 8.

**Sampler.** For MAR variants, we use the ADM (Dhariwal & Nichol, 2021) sampler with 100 steps the same as the original MAR (Li et al., 2024b). For xAR variants, we use the Euler-Maruyama sampler (Ma et al., 2024; Yu et al., 2024d) with 50 steps the same as the original xAR (Ren et al., 2025).

**Computing resources.** We use 16x NVIDIA A100 40GB GPUs for experiments. We use a batch size of 2048 and remain unchanged for all experiments.

**Pseudo-code.** We provide a torch-like pseudo-code of training models with BAR in algorithm 1.

## D    TRAINING WALL-TIME AND EFFICIENCY

### D.1    CONVERGENCE SPEEDUP

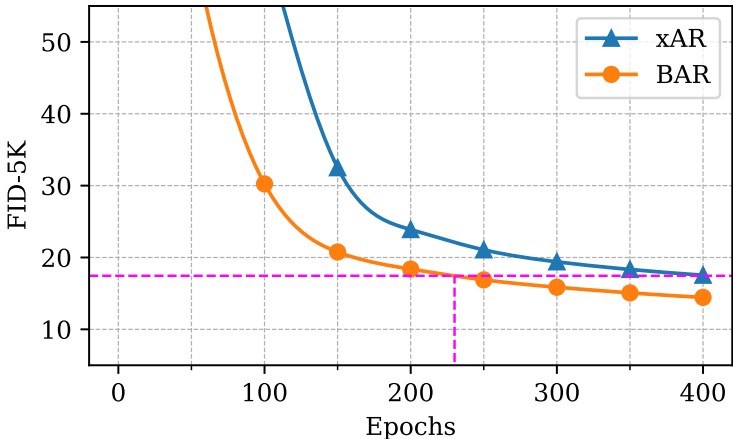

Figure 7: We train BAR-B and xAR-B for 400 epochs. Our BAR-B converges significantly faster than xAR-B, and obtains lower FID-5K throughout the whole training process.

BAR not only improves the final generation quality of AR models, but also accelerates the training process and its convergence, since the image tokens are transformed and refactored to better suit the nature of the next-token prediction paradigm. As illustrated in figs. 3 and 4, the basis and transformed tokens mimic a coarse-to-fine pattern, allowing the AR models to first focus on the global structures then refine the local details.

In fig. 7, we conduct ablation experiment with BAR-B model and compare the intermediate FID-5K evaluation results with xAR-B during the training process. As shown, the convergence speed of BAR is significantly faster than xAR, and our BAR-B model obtains better performance compared to xAR-B under the same iterations throughout the whole training process.

### D.2    TRAINING EFFICIENCY

We also compare the wall-time per iteration and GPU memory usage during training in table 9 using one A100 GPU. Since BAR only introduces an additional matrix multiplication in the training process (see algorithm 1), both the training time and memory cost of BAR is the same as the original xAR. When incorporating the proposed residual loss to explicitly regularize the order within learn basis, the training cost is slightly raised by the overheads of $0.01\times$ wall-time and $0.03\times$ memory.

Table 9: Training efficiency of BAR. Results evaluated on one A100 GPU.

| | xAR | BAR (without residual loss) | BAR (with residual loss) |
|---|---|---|---|
| Wall-time per iteration | $1.00\times$ | $1.00\times$ | $1.01\times$ |
| GPU memory usage | $1.00\times$ | $1.00\times$ | $1.03\times$ |

# E    EXTENSION TO DISCRETE AR MODELS

While this paper predominantly focuses on the exposition and experiments of continuous BAR models, the unified framework of linear transform can also be adapted to discrete AR paradigms. The rationale for prioritizing continuous BAR models herein is that they avoid the quantization of the output introduced by VQ-VAEs. Such quantization, a prerequisite step required by discrete AR models, inherently introduces loss of information. This loss can detrimentally affect the fidelity and quality of the final generated images. Consequently, the adoption of continuous AR models circumvents this quantization-induced information bottleneck, facilitating superior preservation of information directly from the latent representation of VAEs.

There exist two potential strategies to extend BAR to discrete AR models. The first involves adapting the principles of linear transformations and learnable basis to the quantization stage of VQ-VAE. For instance, VAR (Tian et al., 2024) implements a manual designed hierarchical quantization, progressing from global to local image features. This is achieved by resizing the outputs of VAE to various resolutions prior to quantization, effectively dedicating the initial basis to the holistic image representation, subsequent four bases to the quadrants of image, *etc*. In a similar vein, the learned bases, as visualized in fig. 3, derived from a BAR model could potentially guide the quantization of VAE latents. However, it might require a multi-stage training process, or the joint optimization of both VQ-VAE and AR model.

The second relies and utilizes the outputs of existing VQ-VAEs. In this case, the content of each discrete token remains unaltered, and the linear transform is thereby confined to the rearrangement or regrouping of tokens the standard basis $e_i$. The corresponding transform matrix $\mathbf{A}$ takes the form of permutation matrix, which consists of only 0 and 1. This is similar to RAR (Yu et al., 2024b) and PAR (Wang et al., 2024), where the former introduces random permutations of the token sequence during training, and the latter partitions the tokens into groups and prioritizes the pivot tokens. However, this approach may introduce challenges on how to optimize the matrix $\mathbf{A}$ while maintaining it as a permutation matrix.

# F    ABLATION ON RESIDUAL LOSS

Table 10: Compare offline pre-computed basis and online learnable basis with $\mathcal{L}_{\text{residual BAR}}$.

| Model | FID↓ | IS↑ | Pre.↑ | Rec.↑ |
|---|---|---|---|---|
| Baseline | 1.72 | 280.4 | 0.82 | 0.59 |
| Offline basis | 1.69 | 286.4 | 0.82 | 0.60 |
| Online learnable basis | 1.63 | 291.3 | 0.83 | 0.62 |

The design of our proposed $\mathcal{L}_{\text{residual BAR}}$ in eq. (10) naturally leads to the question that can we first apply a similar residual loss on real images from dataset, then fix the offline pre-computed basis to train AR models? We conduct ablation experiments in table 10, and find that while AR models benefit from the offline basis, our online learnable basis allows adjustments to the inherit dynamics of AR models and obtains better performances.

Table 11: $FD_{DINOv2}$, KID, and CLIP score comparison.

| | $FD_{DINOv2}\downarrow$ | KID$\downarrow$ | CLIP score$\uparrow$ |
|---|---|---|---|
| DiT (Peebles & Xie, 2023) | 77.43 | 0.060 | 0.317 |
| SiT (Ma et al., 2024) | 70.52 | 0.055 | 0.326 |
| REPA (Yu et al., 2024d) | 59.16 | 0.052 | 0.334 |
| LightningDiT (Yao & Wang, 2025) | 56.23 | 0.050 | 0.338 |
| BAR | **52.33** | **0.047** | **0.343** |

Table 12: CLIPscore and HPSv2 comparison.

| | CLIP score$\uparrow$ | HPSv2$\uparrow$ |
|---|---|---|
| LDM | 0.330 | 25.56 |
| LlamaGen | 0.319 | 24.22 |
| FAR | 0.328 | 25.28 |
| BAR | **0.332** | **25.71** |

# G MORE RESULTS

## G.1 ADDITIONAL METRICS

To provide a more comprehensive evaluation beyond the commonly utilized Fréchet Inception Distance (FID) (Heusel et al., 2017) and Inception Score (IS) (Salimans et al., 2016), we also incorporate more metrics in table 11, including $FD_{DINOv2}$ (Stein et al., 2023) and Kernel Inception Distance (KID) (Bińkowski et al., 2018). Furthermore, to quantify the alignment between generated images and input conditions, we report the CLIP (Radford et al., 2021) score in tables 11 and 12 by computing the CLIP similarity between images and corresponding prompts. For the class-conditional image generation task on ImageNet, as detailed in table 11, the text prompt is constructed as "a photo of [CLASS]", where [CLASS] is replaced by the specific class label. In the context of text-to-image models, we also present HPSv2 (Wu et al., 2023) results in table 12.

## G.2 LEARNED BASIS VISUALIZATION

In addition to fig. 3, we provide more visualization results of the learned basis in fig. 8. We attribute the discontinuous pattern of latent space FFHQ to the insufficient training, while the basis of latent space ImageNet is optimized for more iterations.

## G.3 GENERATION PROCESS VISUALIZATION

In addition to fig. 4, we provide more visualization results of the generation process by progressively decoding the generated token sequence $\mathbf{y}$ in fig. 9.

## G.4 GENERATED SAMPLES

We present more visualization of generated samples with our BAR-H model in figs. 10 to 17. We also present more text-to-image samples of our BAR model in fig. 18.

FFHQ (pixel space)     FFHQ (latent space)     ImageNet (latent space)

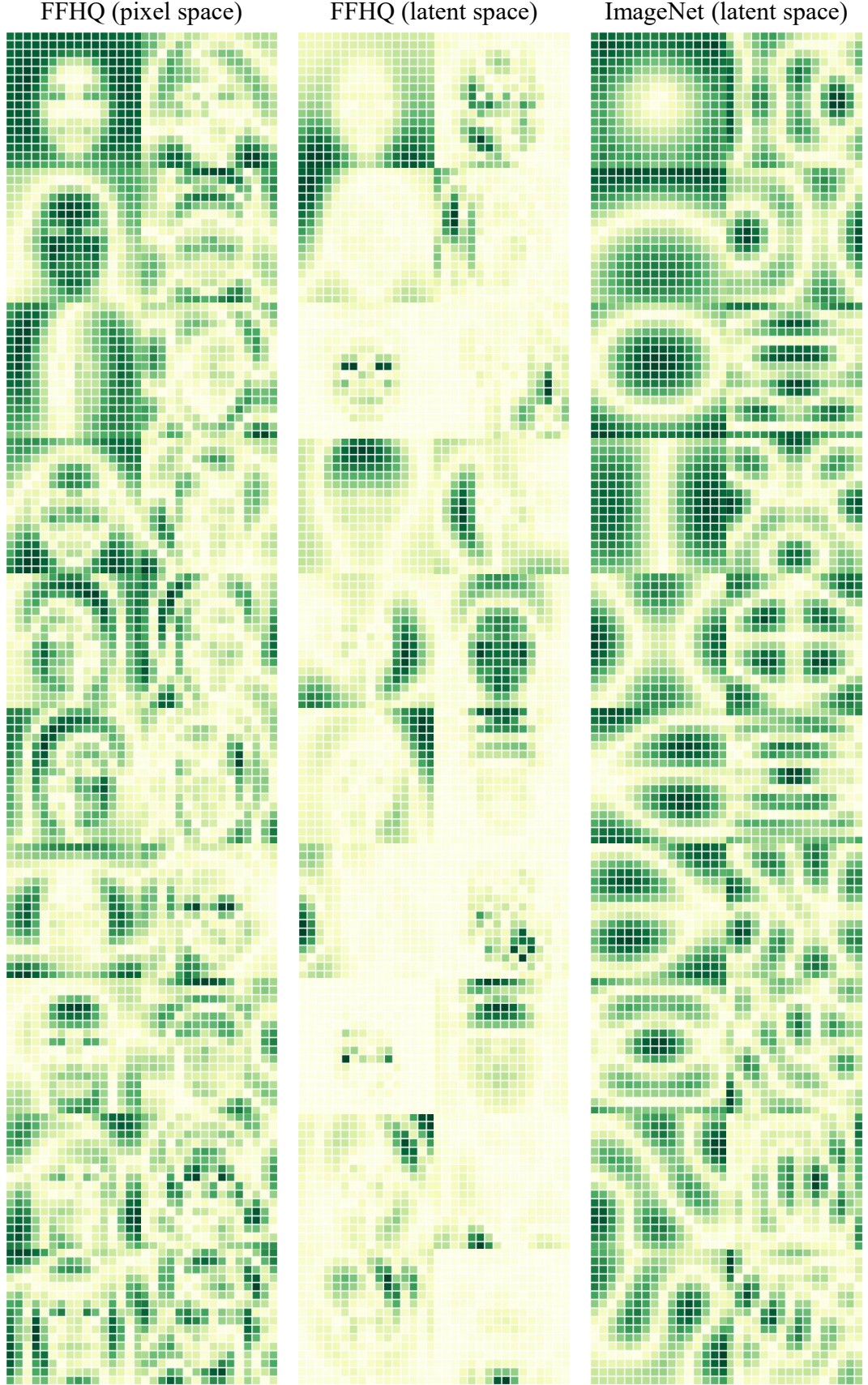

Figure 8: Visualization of the first 20 learned basis $a_k$. Darker regions mark higher values.

FFHQ

ImageNet

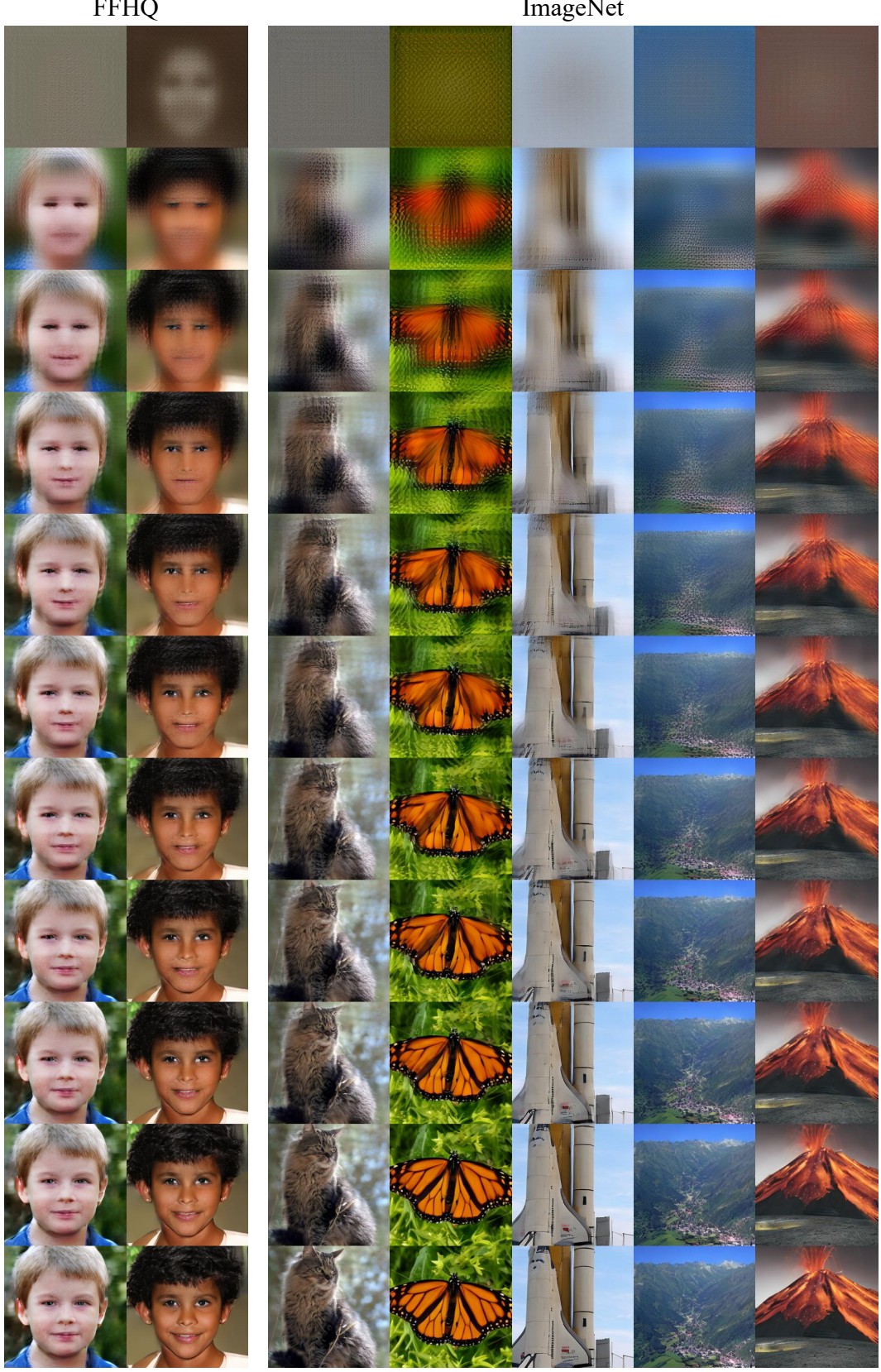

Figure 9: We decode the generated sequences every 25 tokens. The first row uses only one token.

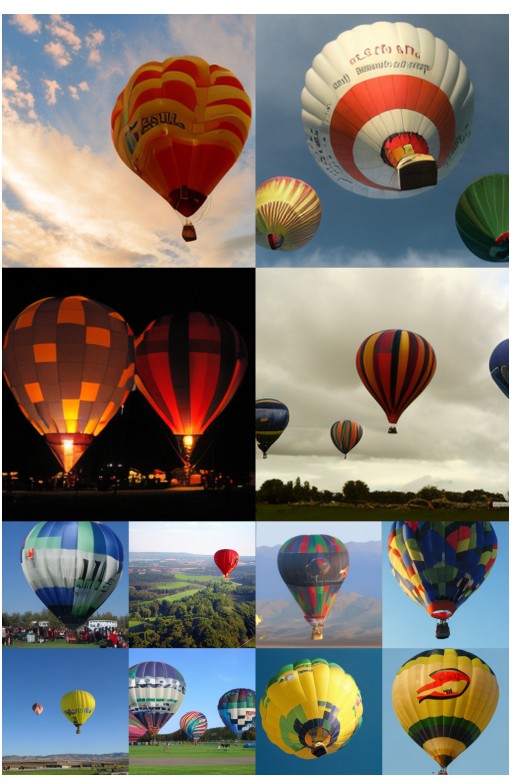

Figure 10: Generated samples of BAR-H on the class "balloon" (417).

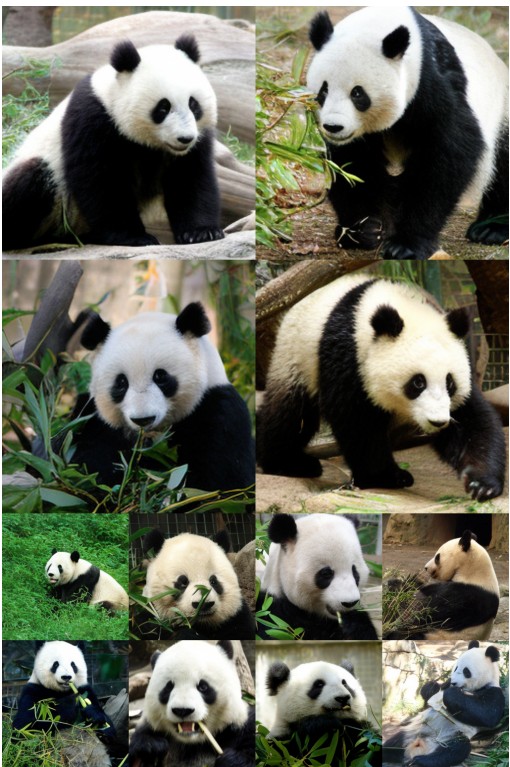

Figure 11: Generated samples of BAR-H on the class "panda" (388).

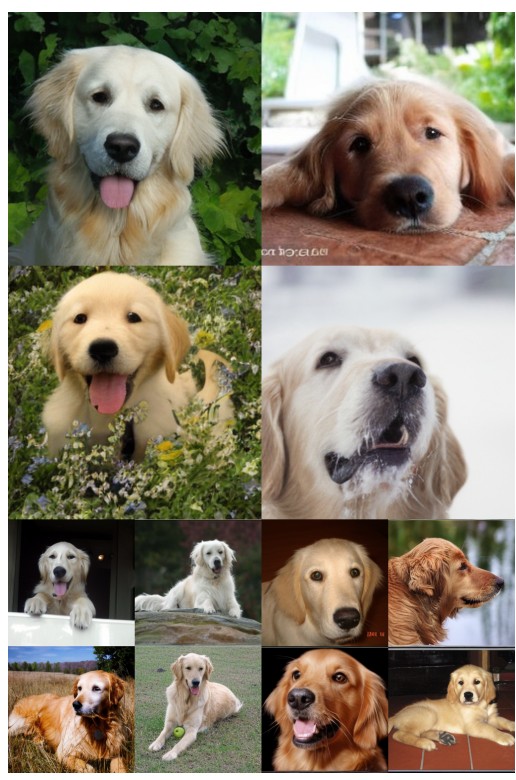 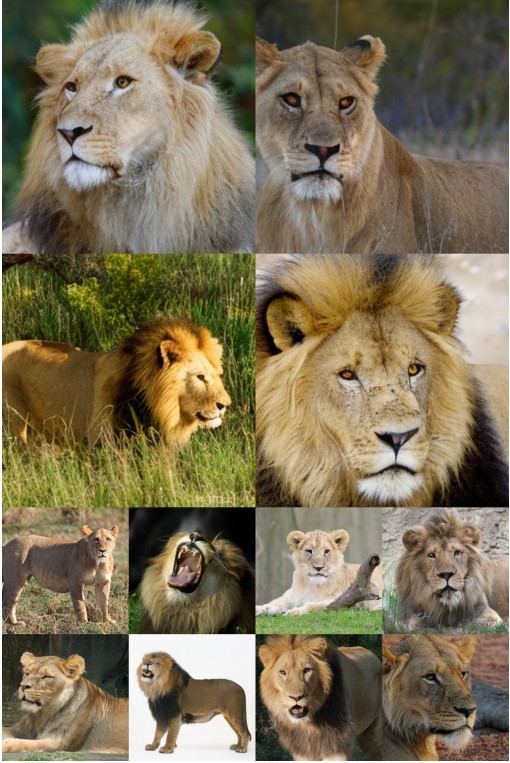

Figure 12: Generated samples of BAR-H on the class "golden retriever" (207).

Figure 13: Generated samples of BAR-H on the class "lion" (291).

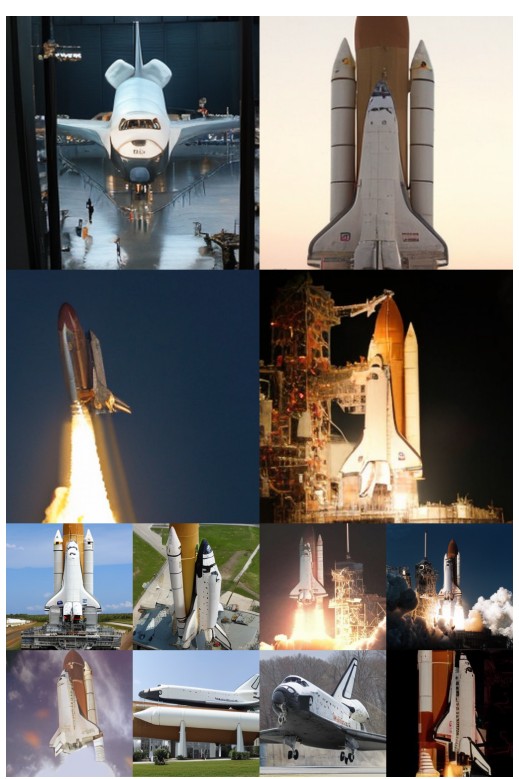

Figure 14: Generated samples of BAR-H on the class "space shuttle" (812).

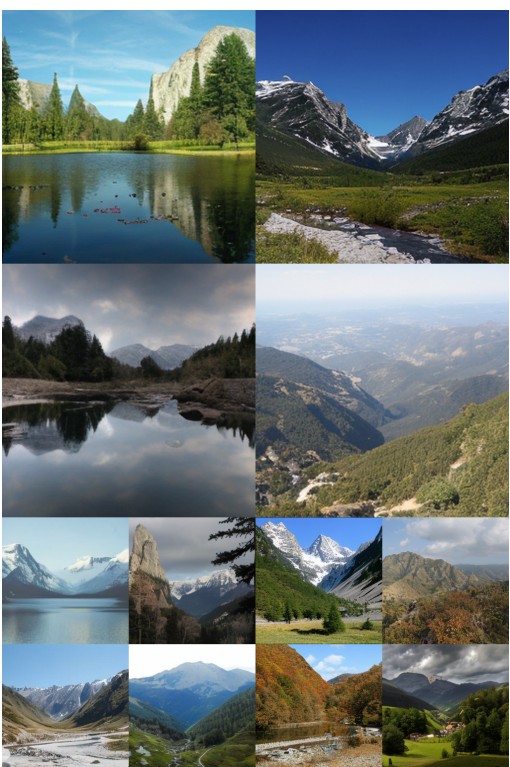

Figure 15: Generated samples of BAR-H on the class "valley" (979).

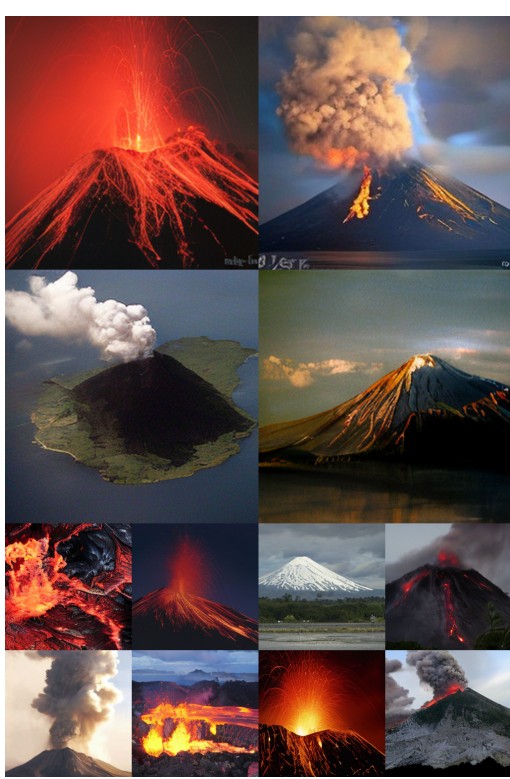 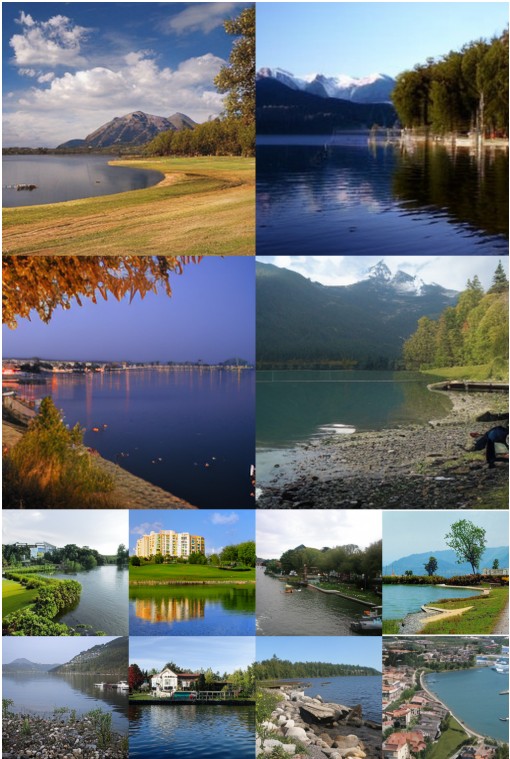

Figure 16: Generated samples of BAR-H on the class "volcano" (980).

Figure 17: Generated samples of BAR-H on the class "lake shore" (975).

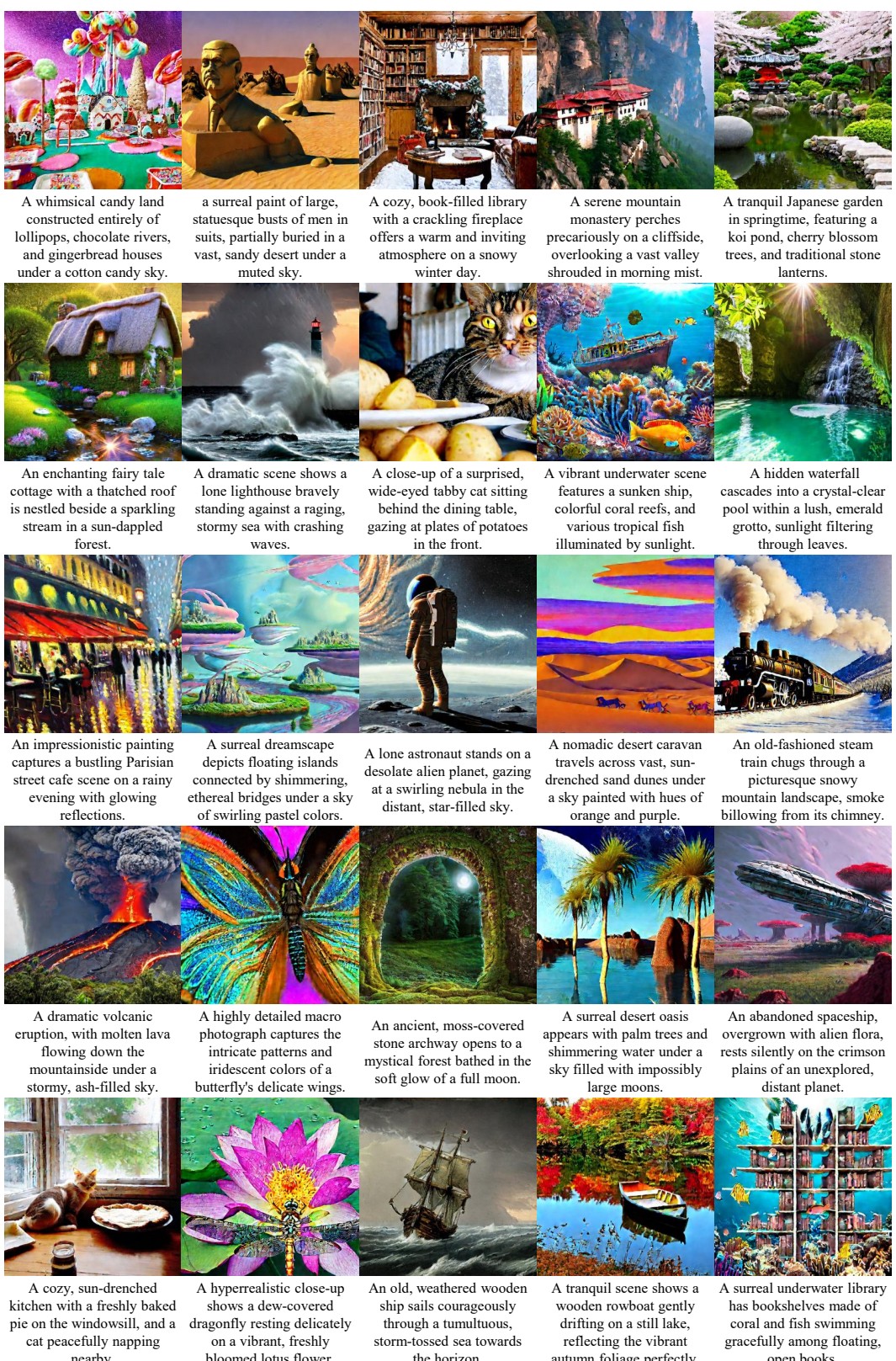

Figure 18: Text-to-image samples of BAR.

