# OpenReview forum: "BAR: Refactor the Basis of Autoregressive Visual Generation"
_ICLR.cc/2026/Conference — ICLR 2026 Poster_

### Official Review · Reviewer_wfTF · 2025-10-22

**Soundness:** 3
**Presentation:** 4
**Contribution:** 4
**Rating:** 6
**Confidence:** 4

**Summary:**

This paper proposes Basis Autoregressive, a unified and learnable framework for autoregressive visual generation that treats tokens as projections of an image onto a learned basis via a linear transform y = Ax. By optimizing an orthogonal, end-to-end trainable transform matrix A jointly with the AR model, BAR refactors token sequences into more informative orders and units, theoretically preserving equivalence to MAR and xAR objectives and practically improving generation. A residual objective encourages early bases to capture maximal information. BAR subsumes prior designs (e.g., VAR, PAR, RAR, FAR, xAR) as special cases and achieves state-of-the-art results on ImageNet-256, with additional results of ImageNet-512, text-to-image generation, and extensive ablations.

**Strengths:**

* The proposed BAR paradigm is fresh and compelling, offering a unified framework that theoretically derives and subsumes a range of recent AR variants.
* The method is simple and end-to-end trainable without complicated designs.
* It achieves strong results on ImageNet-256, comparable to current state-of-the-art diffusion models, and also includes extensive ablations that validate the effectiveness of each component. The paper also validates its scalability in terms of model size, resolution, and data scale (t2i generation).
* The paper is clearly written, well-structured, and easy to follow.

**Weaknesses:**

I do not see any major flaws in this paper. I have a few discussion points, and if addressed, I would be happy to raise my score:

* The residual loss feels somewhat heuristic. The paper’s motivation critiques recent AR work for relying on hand-crafted inductive biases, yet the residual loss reintroduces a prefix-ordering bias. More diagnostics on how the learned ordering correlates with information content, and comparisons to alternative ordering regularizers, would clarify why this particular choice is preferable.

* Orthogonality constraint on A: While enforcing orthogonality yields clean equivalence to MAR/xAR and stabilizes training, it restricts the search space to energy-preserving transforms. It remains unclear whether relaxing to general invertible A could provide further gains. An exploration of non-orthogonal parameterizations would be valuable.

**Questions:**

* Why can A be trained end-to-end jointly with the AR model? What outcomes would we expect from a two-stage training scheme instead?

* In Appendix Figure 18, the quality of text-to-image results does not look good, with many high-frequency artifacts. Please explain the likely causes.

---

> ### Author Response · Authors · 2025-11-25
> **Author Response to Reviewer wfTF (Part 1/2)**
>
> We thank Reviewer wfTF for the positive and encouraging assessment, and for describing BAR as a “fresh and compelling” paradigm with strong results and clear presentation. Here we carefully considered all the points and provide detailed responses below.
>
> ## W1. Residual loss and prefix-ordering bias
>
> First, we would like to clarify our position on inductive bias. Our critique is not *AR should not use inductive biases at all*—inductive biases are crucial for efficient generative modeling, especially in visual domain—but rather how these biases are used, e.g., a flexible and learnable principle rather than rigid, manually designed prediction units and schedules.
>
> Our BAR takes a different stance:
>
> - We introduce a minimal and principled inductive bias only at the level of ordering, while leaving maximum flexibility and let the basis vectors themselves to be fully learned end-to-end from data;
> - Compared with previous methods that directly hand-craft the prediction units and schedules, our residual loss simply encourages earlier basis indices to carry more reconstructive information. This is naturally aligned with (i) the global-to-local structure of visual signals, and (ii) the causal nature of AR prediction, without prescribing what each basis function must look like.
>
> To address the requests on more diagnostics and alternatives, we ran additional experiments with two alternative ordering regularizers:
>
> - Postfix-ordering: encourage later tokens to carry more information (residual losses applied on suffixes rather than prefixes).
> - Random-ordering: sample a random permutation (once sampled it is fixed for each model) and apply the residual objective on its prefixes, enforcing an arbitrary information ordering.
>
> We report the results below:
>
> | Model | FID$\downarrow$ |  IS$\uparrow$ |
> |---|---|---|
> | *Baseline (vanilla BAR without order regulrizer)* | 1.64 | 289.7 |
> | Prefix-ordering | 1.56 | 292.4 |
> | Postfix-ordering | 14.7 | 89.3 |
> | Random-ordering | 6.60 | 192.7 |
>
> We observe that **Prefix-ordering (used in main paper) > no ordering regularizer (vanilla BAR) > Random-ordering > Postfix-ordering** both in terms of FID and convergence speed since that
>
> - Postfix-ordering substantially harms the performance because the model must generate many low-information tokens, like high-frequent details, before it is allowed to determine the global structure, which conflicts with causal prediction.
> - Random-ordering removes the semantic coarse-to-fine progression, and thus leads to unstable prediction for AR across tokens.
>
> These results support our view that the residual loss introduces a well-aligned, minimal inductive bias, rather than an arbitrary heuristic: it imposes just enough structure to harmonize (i) the natural global-to-local organization of images and (ii) the causal next-token prediction mechanism of AR, while still allowing the basis itself to be entirely data-driven.
>
> ## W2. Orthogonality constraint on $\mathbf{A}$ and potential benefits of relaxing it
>
> We respectfully argue that the orthogonality of $\mathbf{A}$ is exactly one of the core designs and contributions of this work. The orthogonality is essential for our framework for the following observations and reasons:
>
> ### 1. Loss and gradient equivalence to MAR/xAR
>
> In Proposition 1, Proposition 2, and Appendix A, we explicitly shown it is only when $\mathbf{A}$ is orthogonal that BAR is equivalent to MAR/xAR.
>
> If we relax orthogonality, $\mathcal{L}\_{\text{BAR}}$ becomes a quadratic form weighted by $\mathbf{A}^\top \mathbf{A}$. The optimization can then "cheat" by reducing $\mathcal{L}\_{\text{BAR}}$ by shrinking the singular values of $\mathbf{A}$ instead of genuinely improving prediction.
>
> This “extra expressivity” in $\mathbf{A}$ is therefore largely a scaling degree of freedom inside $\Sigma$ where $\mathbf{A}=U\Sigma V^\top$, which breaks the equivalence and introduces degenerate solutions, rather than providing a meaningful gain in generative capacity.
>
> ### 2. Diffusion consistency and Gaussian noise geometry
>
> The diffusion head of our BAR follow the standard DDPM-type construction, where the target noise is i.i.d. Gaussian $\varepsilon \sim \mathcal{N}(0,I)$.
>
> In the $\mathbf{A}$-transformed space, the noise becomes $\varepsilon' = \mathbf{A}\varepsilon$ with covariance $\mathrm{Cov}(\varepsilon') = \mathbf{A}\mathrm{Cov}(\varepsilon)\mathbf{A}^\top = \mathbf{A} \mathbf{A}^\top$. Only when $\mathbf{A} \mathbf{A}^\top = I$ do we retain $\varepsilon' \sim \mathcal{N}(0,I)$.
>
> If we allow a invertible but non-orthogonal $\mathbf{A}$, the diffusion head is no longer trained against isotropic noise but against $\mathcal{N}(0, \mathbf{A}\mathbf{A}^\top)$, so the standard DDPM interpretation and the equivalence to MAR/xAR no longer hold without re-deriving the objective under a non-trivial covariance.

---

> ### Author Response · Authors · 2025-11-25
> **Author Response to Reviewer wfTF (Part 2/2)**
>
> ## Q1. Joint training of A vs. two-stage schemes
>
> Conceptually, the “optimal” basis is not determined by data alone—it also depends on the capacity, architecture, and inductive biases of the AR backbone. This motivates joint training:
>
> - When trained end-to-end, $\mathbf{A}$ can adapt to what the transformer finds easier or harder to model. For example, if the Transformer struggles with specific high-frequency structures, $\mathbf{A}$ can rotate those structures into later tokens or redistribute them across basis components in a way that matches the model’s capabilities. One can also imagine that joint training is essentially equivalent to the alternative training of $\mathbf{A}$ and transformer with infinite stages.
> - In a two-stage scheme, e.g. pre-computing $\mathbf{A}$ separately and then freezing it, the basis is fixed and cannot co-adapt with the AR model. In our ablations in Table 10, such two-stage approach yields worse FID and slower convergence compared to joint training. Intuitively, PCA learns a reconstructive basis that maximizes variance, whereas BAR learns a predictive basis optimized specifically for AR prediction under the given backbone.
>
> ## Q2. Text-to-image artifacts in Appendix Figure 18
>
> We agree that some samples in Figure 18 exhibit undesirable high-frequency artifacts, and we appreciate the opportunity to clarify the likely causes below:
>
> ### 1. Sampling: high CFG scale
>
> The samples in Figure 18 were generated with a relatively high classifier-free guidance scale of 4.0, chosen to emphasize text–image alignment and visual quality. It is well known that large CFG scales in latent models can lead to over-saturated colors and ringing-like artifacts. We have confirmed that lowering the CFG scale to 3.0 in consistence to FAR significantly reduces these artifacts.
>
> ### 2. Training dataset: style bias in JourneyDB
>
> Our text-to-image model is trained on JourneyDB, following the same setup as FAR. JourneyDB contains a large fraction of highly stylized, vivid, and sometimes over-sharpened images; as illustrated in the JourneyDB paper [1], e.g., their Table 7 and Figure 5, there are several images exhibit strongly stylized and saturated aesthetics, and the dataset distribution deviates from natural photographs. Consequently, our model inherit part of this stylized visual bias.
>
> [1] Pan, J., Sun, K., Ge, Y., Li, H., Duan, H., Wu, X., Zhang, R., Zhou, A., Qin, Z., Wang, Y., Dai, J., Qiao, Y., & Li, H. (2023). JourneyDB: A Benchmark for Generative Image Understanding. ArXiv, abs/2307.00716.
>
> We sincerely thank you for the helpful suggestions and hope the above response effectively addresses your concerns.

---

> > ### Comment · Reviewer_wfTF · 2025-11-26
> >
> > Thanks for your reply! I am happy to raise my score.

---

### Official Review · Reviewer_WUrW · 2025-10-30

**Soundness:** 3
**Presentation:** 3
**Contribution:** 3
**Rating:** 6
**Confidence:** 4

**Summary:**

The paper proposes Basis Autoregressive (BAR), a novel paradigm that reformulates autoregressive (AR) image generation by viewing token sequences as basis vectors in a linear space. Traditional AR models generate images in raster-scan order, which disregards spatial structure and heavily depends on human-designed heuristics. BAR achieves state-of-the-art FID 1.15 on ImageNet-256, demonstrates scalability to 512×512 resolution, and generalizes to text-to-image tasks with consistent improvements across metric

**Strengths:**

BAR reformulates AR models as linear-space transformations, offering a mathematically elegant and generalizable view that subsumes prior ad hoc approaches.

The introduction of an orthogonal, end-to-end learnable transform matrix A replaces human inductive biases with data-driven optimization.

BAR achieves new SOTA on multiple benchmarks (e.g., FID 1.15 on ImageNet-256) and demonstrates speed–quality advantages with fewer parameters.

**Weaknesses:**

1. While mathematically rigorous, the paper could better explain the intuition behind how the learned basis improves spatial dependency modeling.
2. The exploration of non-orthogonal or adaptive-rank transformations could broaden the framework’s generality.
3. Although efficiency is claimed, a detailed analysis of training overhead (from learning (A)) versus vanilla AR would be valuable.
4.  The connection between BAR’s linear transform and classical linear subspace learning (e.g., PCA, ICA) is briefly implied but not discussed explicitly.
5.  Equation (9)–(10) and residual objective explanation could use more intuitive narrative; some notations (e.g., (ỹ_k)) are underdefined at first appearance.

**Questions:**

1. Does the orthogonality constraint limit the expressivity of (A)? Would relaxing it (e.g., via low-rank factorization) yield additional improvements?
2. How does BAR perform in **long-sequence autoregression** (e.g., 1k+ tokens), where cumulative numerical error in (A^{-1}) may become significant?

---

> ### Author Response · Authors · 2025-11-25
> **Author Response to Reviewer WUrW (Part 1/3)**
>
> We thank Reviewer WUrW for the detailed assessment, and for recognizing BAR as a “mathematically elegant and generalizable” paradigm that achieves SOTA results with speed–quality advantages. We respond to your concerns point by point below.
>
> ## W1. Intuition for how the learned basis improves spatial dependency modeling
>
> We agree that an intuitive view is important. Standard raster-scan AR factorizes $p(x_1, x_2, \dots, x_N)$ along spatial indices and suffers from a blind start: the first token has essentially no context of the global visuals, yet must precisely predict every style, and semantics, and details of the first patch. This causal structure (“top-left $\rightarrow$ bottom-right”) is misaligned with the global-to-local structure of images.
>
> Instead, BAR factorizes in a learned basis $\mathbf{y} = \mathbf{Ax}$ and the probability $p(y_1, y_2, \dots, y_N)$ where each component $y_i = a_i^\top \mathbf{x}$ aggregates information over the entire latent map, and the residual objective orders components by reconstructive importance. Intuitively,
> - Early components become global modes, e.g., low-frequency and large-scale structure, layout, overall color, and style.
> - Later coefficients refine increasingly local, high-frequency details.
>
> Thus the AR causal structure becomes “global structure $\rightarrow$ local details”, which is a much more natural dependency pattern for images and avoids the blind-start issue.
>
> ## W2 & Q1. Non-orthogonal and adaptive-rank / low-rank transforms
>
> We respectfully argue that the orthogonality of $\mathbf{A}$ is exactly one of the core designs and contributions of this work. The orthogonality is essential for our framework for the following observations and reasons:
>
> ### 1. Loss and gradient equivalence to MAR/xAR
>
> In Proposition 1, Proposition 2, and Appendix A, we explicitly shown it is only when $\mathbf{A}$ is orthogonal that BAR is equivalent to MAR/xAR.
>
> If we relax orthogonality, $\mathcal{L}\_{\text{BAR}}$ becomes a quadratic form weighted by $\mathbf{A}^\top \mathbf{A}$. The optimization can then "cheat" by reducing $\mathcal{L}\_{\text{BAR}}$ by shrinking the singular values of $\mathbf{A}$ instead of genuinely improving prediction.
>
> This “extra expressivity” in $\mathbf{A}$ is therefore largely a scaling degree of freedom inside $\Sigma$ where $\mathbf{A}=U\Sigma V^\top$, which breaks the equivalence and introduces degenerate solutions, rather than providing a meaningful gain in generative capacity.
>
> ### 2. Expressivity of orthogonal $\mathbf{A}$
>
> The above and the analysis in the main paper (Sec.3, Appendix A) shows that, with orthogonal $\mathbf{A}$, the BAR objective is exactly equivalent to the original MAR/xAR objective, and both the loss and its gradient are preserved under the change of basis. In other words, **BAR with orthogonal $\mathbf{A}$ does not reduce the expressive capacity**; it represents the same underlying class of functions as MAR/xAR, parameterized in a different coordinate system. The expressivity of BAR is therefore identical to that of MAR/xAR, while the basis refactorization changes the factorization structure and optimization dynamics.
>
> ### 3. Diffusion consistency and Gaussian noise geometry
>
> The diffusion head of our BAR follow the standard DDPM-type construction, where the target noise is i.i.d. Gaussian $\varepsilon \sim \mathcal{N}(0,I)$.
>
> In the $\mathbf{A}$-transformed space, the noise becomes $\varepsilon' = \mathbf{A}\varepsilon$ with covariance $\mathrm{Cov}(\varepsilon') = \mathbf{A}\mathrm{Cov}(\varepsilon)\mathbf{A}^\top = \mathbf{A} \mathbf{A}^\top$. Only when $\mathbf{A} \mathbf{A}^\top = I$ do we retain $\varepsilon' \sim \mathcal{N}(0,I)$.
>
> If we allow a invertible but non-orthogonal $\mathbf{A}$, the diffusion head is no longer trained against isotropic noise but against $\mathcal{N}(0, \mathbf{A}\mathbf{A}^\top)$, so the standard DDPM interpretation and the equivalence to MAR/xAR no longer hold without re-deriving the objective under a non-trivial covariance.
>
> ### 4. Non-full-rank transforms imply information loss
>
> If we use a adaptive-rank or low-rank $\mathbf{A} \in \mathbb{R}^{N\times N}$ with $\mathrm{rank}(A)=r<N$, instead of the full-rank ones, we have $\dim\ker(A)=N-r>0$. Hence $\mathbf{A}$ is non-invertible on $\mathbb{R}^N$, and $\mathbf{x} \mapsto \mathbf{y} = \mathbf{Ax}$ is a compression onto an r-dimensional subspace rather than a refacforization of the same latent space. This violates the invertibility assumption underlying the equivalence of BAR and changes the problem from “refactorizing the basis of the same latent space” to “projecting onto a lower-dimensional subspace”, which is outside the scope of and can in turns harm the expressivity of the BAR method.

---

> ### Author Response · Authors · 2025-11-25
> **Author Response to Reviewer WUrW (Part 2/3)**
>
> ## W3. Training overhead from learning $\mathbf{A}$
>
> In Appendix D.2 and Table 9, we have already measured training overhead in terms of both per-step wall-clock time and GPU memory usage of learning $\mathbf{A}$ and residual loss:
>
> - For the learning of $\mathbf{A}$, the overhead is indistinguishable and less then $0.5$% (see the *BAR without residual loss* column);
> - For the residual loss, the overhead is merely $1$% on per-step wall-clock time and $3$% on GPU memory usage (see the *BAR with residual loss* column);
> - This matches the implementation: we add only few matrix multiplies with $\mathbf{A}$ plus an orthogonality regularizer on top of the main transformer model.
>
> ## W4. Connection to PCA / ICA and classical subspace methods
>
> We agree that BAR can be considered as related to classical linear subspace methods, but it optimizes a different objective and plays a different role from them:
>
> ### 1. Relation to PCA
>
> PCA learns an orthogonal basis $U$ that diagonalizes the covariance and orders components by explained variance, i.e., it optimizes a reconstructive criterion. In contrast, BAR learns an orthogonal basis $\mathbf{A}$ jointly with a nonlinear AR backbone under a MAR/xAR-style diffusion objective plus a residual ordering loss. The basis is therefore optimized for predictive performance in stead of variance. Empirically, replacing $\mathbf{A}$ with a fixed PCA basis yields worse FID and slower convergence in Table 10 in the paper, which confirms that a PCA basis is not optimal for the AR generation.
>
> ### 2. Relation to ICA.
>
> ICA seeks a demixing matrix $W$ such that $s = Wx$ has approximately independent components, often by maximizing non-Gaussianity. Even when implemented on whitened data (so $W$ can be orthogonal in the whitened space), the target is independence, not reconstruction or predictive likelihood. BAR does not impose independence on $\mathbf{y}$. Instead, it explicitly models dependencies between components via the AR transformer, i.e. we want the latter components to be relied on former ones in order to suit the causal prediction of AR. In conclusion, ICA attempts to remove dependencies, whereas BAR restructures them along a factorization that is suit to AR.
>
> ## W5. Clarity of Equations (9)–(10) and the explanation of notations ($\tilde{y}_k$)
>
> We agree that a more intuitive narrative, in addition to the mathematical formulation in Equations (9)–(10) and Sec. 3.4, can be helpful for readers to understand our method. We also admit that some notations, including $\tilde{y}_k$ while defined in Sec. 3.4, should be explained at the first appearance.
>
> Intuitively, the residual objective is encouraging the following behavior: if we only keep the first $k$ tokens of the generated sequence $\mathbf{y}$ from BAR and drop the remaining tokens, the image reconstructed from these $k$ tokens should approximate the final image as close as possible. In other words, early tokens are encouraged to summarize global and dominating structure (layout, color, shapes), and later tokens are used mainly to refine details. This is exactly the coarse-to-fine ordering we want for an AR sequence to possess for visual signals. We will also add this explanation for the notation $\tilde{y}_k$ right in the caption of Figure 2.

---

> ### Author Response · Authors · 2025-11-25
> **Author Response to Reviewer WUrW (Part 3/3)**
>
> ## Q2. Long-sequence autoregression and numerical stability of $\mathbf{A}^{-1}$
>
> In our experiments, the longest sequences arise from ImageNet-512 with a KL-16 tokenizer, yielding a $32\times 32$ latent grid and $N = 1024$ tokens. We firstly want to clarify that since we restrict $\mathbf{A}$ to be orthogonal, **we assume and directly use $\mathbf{A}^{-1}:=\mathbf{A}^\top$, and never actually calculate $\mathbf{A}^{-1}$**. We then discuss the numerical stability at this scale of $N = 1024$:
>
> ### 1. Ideal orthogonality
>
> In the ideal case where $\mathbf{A}$ is exactly orthogonal, we have $\mathbf{A}^{-1}$ = $\mathbf{A}^\top$ and condition number $\kappa(\mathbf{A}) = 1$. Decoding uses a single multiplication by $\mathbf{A}^\top$, so there is no iterative inversion and no accumulation of numerical error across steps.
>
> ### 2. Approximate orthogonality in practice
>
> In practice, $\mathbf{A}$ is enforced to be only **approximately** orthogonal via a loss regularizer and soft projections, but the singular values remain tightly clustered around 1. We can write the SVD decomposition as $\mathbf{A} = U\Sigma V^\top$, where $\Sigma = \mathrm{diag}(\sigma_i)$. We have
>
> $$
> \mathbf{A}^\top - \mathbf{A}^{-1} = V(\Sigma - \Sigma^{-1})U^\top,
> $$
>
> and hence
>
> $$
> \|\mathbf{A}^\top - \mathbf{A}^{-1}\|_2 = \|\Sigma - \Sigma^{-1}\|_2 = \max_i \|\sigma_i - \sigma_i^{-1}\|.
> $$
>
> If all singular values are bounded closely around 1 and satisfy $\sigma_i \in [1-\varepsilon, 1+\varepsilon]$ for some $0 < \varepsilon < 1$, a simple calculation shows that the worst case is attained at $\sigma_j = 1-\varepsilon$ for some $j$ and
>
> $$
> \max_{\sigma_i \in [1-\varepsilon,1+\varepsilon]} \|\sigma_i - \sigma_i^{-1}\|
> = \frac{2\varepsilon - \varepsilon^2}{1-\varepsilon}
> = 2\varepsilon + O(\varepsilon^2).
> $$
>
> In particular, for $\varepsilon \le 1/2$ we obtain the explicit bound
>
> $$
> \|\mathbf{A}^\top - \mathbf{A}^{-1}\|_2
> \le \frac{2\varepsilon - \varepsilon^2}{1-\varepsilon}
> \le 4\varepsilon.
> $$
>
> Thus, as long as the singular values $\sigma_i$ stay within a narrow band around 1 (which is exactly what our soft orthogonality regularizer enforces), $\mathbf{A}$ remains extremely well-conditioned and $\mathbf{A}^\top$ is an $O(\varepsilon)$-accurate approximation of the true inverse. This confirms that the inversion step is numerically stable even for long sequences, and the approximate orthogonality does not pose a practical issue for BAR.
>
> We have also run experiments with the KL-8 tokenizer on ImageNet-512 that corresponds to the token length of $N = 4096$. We report the experiment results below
>
> | Model | FID$\downarrow$ | IS$\uparrow$ |
> |---|---|---|
> | xAR | 1.64 | 290.1 |
> | BAR | 1.60 | 298.4 |
>
> Empirically, at both $N = 1024$ and $N = 4096$ we do not observe any instability or degradation that could be attributed to numerical issues in $\mathbf{A}^{-1}$; BAR consistently improves over the baseline.
>
> We hope the above response clarifies the concerned points. And we thank you again for your constructive evaluation.

---

### Official Review · Reviewer_e8ty · 2025-11-01

**Soundness:** 3
**Presentation:** 3
**Contribution:** 3
**Rating:** 6
**Confidence:** 3

**Summary:**

This paper proposes BAR (Basis Autoregressive), a new paradigm for AR visual generation. Instead of following a fixed raster-scan token prediction order, BAR introduces a learnable linear transform matrix A that redefines the basis of token prediction space. The authors claim that existing AR variants (e.g., VAR, RAR, PAR, FAR, xAR) can be expressed as special cases of A. They further propose a residual training objective to encourage informative early tokens and show extensive experiments achieving FID 1.15 on ImageNet-256, surpassing diffusion and AR baselines. The method is compatible with both MAR and xAR architectures and improves efficiency while reducing inductive bias.

**Strengths:**

1. Unified theoretical framework: Reformulates a broad range of AR improvements under a single matrix transformation paradigm, offering conceptual clarity.
2. Learnable basis with end-to-end training reduces reliance on handcrafted heuristics and inductive biases.
3. Strong empirical results: Achieves new SOTA on ImageNet-256 and competitive performance on ImageNet-512 and text-to-image tasks.
4. Compatible with multiple AR architectures (MAR, xAR), indicating generality rather than architecture-specific tweaking.
5. Residual objective design mimics coarse-to-fine prediction in a principled way rather than manually defining scales.
6. Visualization of learned basis provides interpretability insights into the emergence of hierarchical token prediction.

**Weaknesses:**

1. **Theoretical depth vs. practical benefit**: While the “unified” perspective is appealing, the core operation (learnable linear transform on tokens) is conceptually simple. The novelty may be perceived as incremental unless the generality claim is further formalized or proven beyond examples.
2. **Orthogonality constraint and optimization stability**: The reliance on orthogonal projection raises the question of whether the learned A collapses to simple permutations or low-rank patterns. More analysis is needed.
3. **Lack of ablation on scalability of A**: The matrix is N×N, which scales quadratically with sequence length. It's unclear how this approach behaves at 1024+ tokens or 1024² resolution.
4. **Training cost overhead**: The paper reports inference-time speedup but does not clearly quantify the extra training overhead introduced by optimizing A.

**Questions:**

1. Does A converge to a stable configuration, or does it keep drifting during training? Any visualization of A evolution?
2. Can BAR be applied to autoregressive text LLMs? If not, what limits the transfer?

---

> ### Author Response · Authors · 2025-11-25
> **Author Response to Reviewer e8ty (Part 1/2)**
>
> We thank Reviewer e8ty for the constructive review and for highlighting our “strong empirical results” and “unified theoretical framework.” Below we would like to address and respond to your concerns.
>
> ## W1. Theoretical depth vs. practical benefit
>
> We agree with you that the core operation of our method is conceptually simple. However, it is supported by the underlying substantial mathematical insights and bedrocks, as in Sec.3 and Appendix.A, and we see this simpleness yet effectiveness as compelling strength of our method.
>
> **From next-token to next-basis prediction**
>
> Standard AR assumes a fixed, human-designed factorization $p(x_i \mid x_{<i})$, e.g., raster, random, or scale-based orders. BAR instead learns a basis $\mathbf{A}$ and models $\mathbf{y=Ax}$ and $p(y_i \mid y_{<i})$, so it can better suit to the causal nature of AR models.
>
> **Unified view of existing AR variants**
>
> Under this linear-basis perspective, many recent visual AR variants are corresponded to different certain choices of $\mathbf{A}$. While these prior works have shown significant improvements and are proved not incremental, BAR subsumes these as special cases and further replaces manual design with a single learnable, end-to-end trained transform.
>
> **Simplicity as a strength**
>
> We see the algebraic simplicity of $\mathbf{y=Ax}$ as an core advantage instead of limitation. Because the change to existing AR architectures is minimal (one linear transform and a residual objective), BAR is easy to implement, combine, and extend on top of current and future AR codebases. In that sense it is intentionally “simple but useful”: it provides a principled reparameterization with clear theory, while keeping the barrier to adoption very low.
>
> **Practical gains with minimal modifications**
>
> Empirically, BAR achieves strong improvements, e.g., FID 1.15 on ImageNet-256, competitive ImageNet-512 and text-to-image results, over other specialized designs while requiring only lightweight changes.
>
> ## W2. Orthogonality constraint and potential collapse of $\mathbf{A}$
>
> As we reported in the main paper, we do not observe collapse of $\mathbf{A}$ to permutations or low-rank-like degenerate patterns.
>
> **Visualizations of Basis structure**
>
> In Figure.3 and Figure.9, visualizations of learned basis vectors show that early components capture global, low-frequency, scene-level structures, e.g., layout, large color blocks, coarse shapes, while later components specialize in localized, high-frequency details. This behavior is obviously not simple permutations or trivial block-structured patterns.
>
> **Regularization of orthogonality**
>
> We enforce orthogonality via a loss regularizer on $\mathbf{A}^\top\mathbf{A} - I$ combined with occasional soft projections. This keeps $\mathbf{A}$ closely on the Stiefel manifold. Orthogonality means that $\mathbf{A}$ is always a full-rank matrix and cannot collapse to low-rank patterns.
>
> ## W3. Scalability of $\mathbf{A}$ as an $N\times N$ matrix
>
> In our experiments in the main paper, we have validated the sequence up to $N = 1024$ tokens on ImageNet-512 with a KL-16 tokenizer. Here we have also run experiments on ImageNet-512 with the KL-8 tokenizer that corresponds to the token length of $N = 4096$, and report the results below
>
> | Model | FID$\downarrow$ | IS$\uparrow$ |
> |---|---|---|
> | xAR | 1.64 | 290.1 |
> | BAR | 1.60 | 298.4 |
>
> At both $N = 1024$ and $N = 4096$ scales:
>
> - Parameter count: $\mathbf{A}$ has $N^2$ parameters, that is $\approx65$K for $N=256$, $\approx1$M for $N=1024$, and $\approx16.8$M for $N=4096$, and is small compared to hundreds of millions and even billions of parameters in the Transformer backbone.
>
> - Compute: Applying $\mathbf{y=Ax}$ costs $O(N^2)$ operations, the same asymptotic complexity as self-attention. In practice, the runtime contribution of this single matrix multiplication per forward pass is minor relative to the multi-layer attention and MLP blocks.
>
> - Stability and performance: Empirically, at both scale we do not observe any instability or degradation issues related to $\mathbf{A}$; BAR consistently improves over the baseline.

---

> ### Author Response · Authors · 2025-11-25
> **Author Response to Reviewer e8ty (Part 2/2)**
>
> ## W4. Training cost overhead
>
> We would like to clarify that Table 9 in the paper is precisely reporting the **training overhead** of BAR, with results evaluated on a single A100 GPU.
>
> **What Table 9 shows**
>
> Table 9 reports per-iteration training wall-time and GPU memory usage for BAR versus the corresponding xAR baselines on ImageNet-256, measured on one A100 GPU. Across the configurations we report, BAR stays within $1.00\times\sim1.03\times$ of the baseline in both runtime and memory; the cost and overhead are negligible.
>
> **Interpretation**
>
> This matches the implementation: optimizing $\mathbf{A}$ adds only few $N \times N$ matrix multiplications plus a small orthogonality regularizer, which are inexpensive compared to the multi-layer Transformer stack. In other words, BAR does not introduce a meaningful training overhead beyond the baseline AR model, as quantified in Table 9.
>
> ## Q1. Convergence behavior of A and its evolution
>
> We observe that A stabilizes during the training process:
>
> - We track $\|\mathbf{A}\_t - \mathbf{A}\_{t-m}\|_F$ and $\|\mathbf{A}\_t - \mathbf{A}\_{t-m}\|_2$ over iterations $t$ with the interval $m=1000$, this quantity drops quickly in the initial phase and then becomes small compared to the backbone’s parameter updates, indicating that the basis converges and changes only gradually afterward.
> - We also print visualizations of the basis at multiple checkpoints and they show the similar pattern: global-to-local structure emerges early and then refines, rather than continuously reshuffling or shifting.
>
> In the revision, we will (i) add the convergence plots of $\|\mathbf{A}\_t - \mathbf{A}\_{t-m}\|_F$ and $\|\mathbf{A}\_t - \mathbf{A}\_{t-m}\|_2$ over training and (ii) include basis snapshots at several training iterations to directly illustrate the evolution and stabilization of $\mathbf{A}$.
>
> ## Q2. Applicability of BAR to autoregressive text LLMs
>
> Conceptually, the idea of “next-basis prediction” is not restricted to images, but practical obstacles arise in the text setting:
>
> ### Discrete vs. continuous representations.
>
> BAR as presented operates on continuous image latents, where linear combinations $\mathbf{y=Ax}$ are meaningful. For text, the primary objects are discrete token indexes. One could in principle apply BAR to continuous embeddings of wordpiece and then perform basis-space predictions, but this requires designing a consistent decoding mechanis, may introduce additional complexity, and is beyond our area of ​​expertise.
>
> ### Ordering and structure.
>
> If one wants to preserve the tokens of text to be discrete indexes, $\mathbf{A}$ may need to be further constrained, for example, to be permutation matrixes. This turns the learning of $\mathbf{A}$ into a discrete or combinatorial problem that potentially requires RL or other non-differentiable techniques. This is a different design space from the continuous, spatially structured image latents we focus on.
>
> We therefore see BAR for text LLMs as an interesting but separate research direction. We focus on visual latent spaces in this paper, and will make this limitation and potential extension more explicit in the discussion of future work.
>
> We hope this point-wise response clarifies both the theoretical and practical aspects of BAR and addresses your concerns.

---

### Official Review · Reviewer_shV6 · 2025-11-01

**Soundness:** 3
**Presentation:** 2
**Contribution:** 3
**Rating:** 4
**Confidence:** 3

**Summary:**

This paper introduces Basis Autoregressive (BAR), a framework for visual autoregressive (AR) generative modeling that rethinks how image data is factorized for sequential prediction. Rather than adhering to the fixed, human-designed raster-scan order, BAR mathematically formalizes token sequences as projections onto learned basis vectors, proposing an end-to-end approach to optimize this basis using a parameterized linear transformation matrix $\mathbf{A}$. The framework unifies various previous AR methods as special cases of matrix $\mathbf{A}$ and claims to transcend the limitations of hand-crafted inductive biases, showing strong empirical results (notably FID of 1.15 on ImageNet-256). The paper provides theoretical justification, thorough ablations of the learned basis, and visualizations of both the basis and progressive generation.

**Strengths:**

1. Unified and General Framework: BAR provides a mathematically coherent and unifying lens for understanding and extending autoregressive visual models. By interpreting previous methods as instances of linear basis transformations, the paper brings much-needed formalism to a field heavily reliant on heuristics.
2. Mathematical Depth and Rigor: The paper gives detailed derivations and proofs (see Section 3.3, and Appendix A) demonstrating the theoretical validity and equivalence of BAR to well-known AR objectives (e.g., MAR, xAR), subject to the choice of loss and basis.
3. Visualization: Visualization of generation (Figures 4, 8, 9) and qualitative samples (Figures 5, 10, 11) effectively demonstrate the dynamics and diversity of the BAR approach.

**Weaknesses:**

1. Experimental Reproducibility Gaps: While the main algorithm and hyperparameters are explained (see Section C and Table 8), some necessary details for replication are spread across appendices and main text, and others (e.g., code availability, certain training curves beyond ImageNet, data preprocessing for alternative datasets) are left somewhat vague. For a contribution at this level, full transparency would further strengthen the impact.
2. Dependence on VAE/Tokenizer Quality: Since the BAR method operates atop latent tokens produced via VAE or similar encoders (e.g., KL-16 tokenizer), its success is inextricably tied to the information bottleneck of the encoder. While this is noted in the limitations, reliance on a potentially lossy front-end may restrict the impact in domains where end-to-end optimization is required.
3. Clarity and Structure Issues: The mathematical notation is dense and sometimes overly compressed. For example, the definitions of prefix sums and the role of the reverse transform $\mathbf{A}^{-1}$ in generation/decoding steps are not always explicitly illustrated or described in algorithmic terms (see Figure 2 and the pseudo-code).
Some implementation choices, such as hard vs. soft orthogonal projection, are only detailed in ablation (Table 6) and could use a deeper explanation in the main text.

**Questions:**

Please provide responses to the issues raised in my Weaknesses section.

---

> ### Author Response · Authors · 2025-11-25
> **Author Response to Reviewer shV6 (Part 1/2)**
>
> We thank Reviewer shV6 for the thoughtful assessment and for recognizing the “mathematical depth and rigor” and the “unified and general framework” of BAR, as well as the usefulness of our visualizations. We respond to your three Weaknesses (W1–W3) point by point below.
>
> ## W1. Experimental reproducibility and transparency
>
> We fully share your emphasis on reproducibility and transparency. Our current settings and release plan are as follows.
>
> ### Data preprocessing
>
> For ImageNet-256/512, we strictly follow the preprocessing protocol in MAR and xAR (same resolution, only horizontal flipping, no other augmentation). For FFHQ, we adopt exactly the same preprocessing as on ImageNet. For JourneyDB in the text-to-image experiments, we also strictly follow the FAR setup.
>
> ### Code and training curves beyond ImageNet-256
>
> We understand the request for more training diagnostics and code release across all datasets. However, at the moment, there are two practical constraints that prevent us from immediately releasing full code, checkpoints, and logs for every setting:
>
> - Internal policy: our current internal process requires a formal review before releasing complete logs (e.g., per-epoch metrics, full Weights & Biases runs) and large model checkpoints.
>
> - Data-/model-related risk on non-ImageNet datasets: for datasets involving faces or potentially sensitive content, especially FFHQ, releasing detailed training artifacts entails additional legal and ethical considerations. We aim to comply with major venue ethics guidelines and to avoid releasing models or logs that could inadvertently increase the risk of misuse.
>
> Within these constraints, we have already started ongoing the open-sourcing procedure for BAR, including releasing the code implementation, model checkpoints, associated logs, and full Weights & Biases docs where permitted. We deeply apologize for any possible inconveniences here and will make every efforts to ensure the reproducibility and transparency of this work.
>
> ## W2. Dependence on VAE / tokenizer quality
>
> We agree that BAR, like latent diffusion and latent AR models in general, inherits an information bottleneck from the tokenizer. However, we view this as a shared constraint of latent generative modeling rather than a limitation specific to BAR.
>
> ### Same VAE, better latent modeling
>
> All baselines we compare against (xAR and MAR) use the same VAE / tokenizer as BAR. The key difference is in how the latent sequence is factorized and modeled. By achieving substantially improved FID, e.g., 1.15 on ImageNet-256, under identical tokenizers and training data, BAR demonstrates that it models the latent distribution more accurately, not that it uses a different front-end.
>
> ### Drift vs. reconstruction
>
> In practice, artifacts in latent generative models often arise not because the VAE cannot reconstruct its own prior, but because the generative model produces out-of-distribution latents (“drift”) that fall outside the VAE’s training manifold. The decoder then receives latent codes it never saw during training. BAR’s global-first factorization and residual objective reduce this drift by:
> - predicting global basis coefficients early, which capture coherent global structure of the latent map;
> - encouraging early components to be maximally reconstructive and allow latter components to rectify errors, keeping generated latents closer to the VAE’s valid manifold.
>
> ### End-to-end vs. modular design
>
> Our framework is orthogonal to the choice of tokenizer: if a stronger or end-to-end latent representation becomes available, BAR can be applied on top of it in exactly the same way. BAR addresses the factorization and modeling of a given latent space, rather than the design of the encoder itself.

---

> ### Author Response · Authors · 2025-11-25
> **Author Response to Reviewer shV6 (Part 2/2)**
>
> ## W3. Clarity and structure of mathematical notation and implementation details
>
> We appreciate the comment that the notations in Sec. 3 and Appendix is overly compact and dense to some degree since we want to deliver a fully rigorous and self-contained derivation of BAR’s equivalence to MAR/xAR and its diffusion formulation. We also agree that some additional intuitive explanations and earlier clarifications would make the paper easier to follow.
>
> ### The prefix sums
>
> The “prefix sum” here can be read as: “only keep the first $k$ tokens of the generated sequence $\mathbf{y}$ from BAR and drop the remaining tokens by setting them to be zero.” Intuitively, the prefix sums are just “partial generations”; incrementing $\tilde{y}_k$ is exactly a coarse-to-fine generation for visual signals. Therefore we can encourage early tokens are encouraged to summarize global and dominating structure (layout, color, shapes), and later tokens are used mainly to refine details.
>
> ### The reverse transform
>
> Conceptually, BAR works in two coordinate systems: $\mathbf{x}$ in the original latent space and $\mathbf{y}$ in the $\mathbf{A}$-transformed space. The forward transform $\mathbf{y=Ax}$ is used during training to define the sequence that the AR model predicts. The reverse transform $\mathbf{x=A^\top y}$ is how we come back to the original latent space before sending to decoder. Intuitively, one can think of it as: “The AR model draws the picture in its own preferred basis; the reverse transform converts that back into the standard latent image that the VAE decoder understands.”
>
> ### Hard and soft orthogonal projection
>
> Orthogonality is how we keep $\mathbf{A}$ as a pure rotation/reflection of the latent space, without scaling or distortion. Beside loss regularizers, we develop two ways to enforce this:
>
> - Hard projection: we forcibly project $\mathbf{A}$ back onto the set of exactly orthogonal matrices. Intuitively, this keeps $\mathbf{A}$ perfectly on the “orthogonal surface” at all times, but also means the optimizer is not allowed to take any shortcut through nearby non-orthogonal directions.
>
> - Soft projection: we instead allow $\mathbf{A}$ to be closely orthogonal but not need to be perfectly orthogonal. Intuitively, this lets $\mathbf{A}$ wander a bit off the surface when it helps optimization, while gently pulling it back towards orthogonality.
>
> Our ablations show that the soft version works better in practice: it still keeps $\mathbf{A}$ very close to orthogonal so the theory and stability benefits remain, and also avoids over-constraining the optimization trajectory.
>
> We hope these clarifications address your concerns about reproducibility, tokenizer dependence, and clarity, and we thank you again for your careful and constructive review.

---

### Author Response · Authors · 2025-12-03
**General Response and Rebuttal Summary**

Dear PCs, SACs, ACs, and Reviewers,

We would like to express our gratitude for the time and effort you have devoted to overseeing the review process for our submission. We deeply appreciate the difficult job of managing reviews under the current extraordinary circumstances, so we would like to provide a minimal summary for your convenience.

Firstly, our work proposes a learnable end-to-end autoregressive image generation model that transcend beyond manually designed prediction units and fixed pattern choices. By doing so, we overcome the fundamental limitations of prior methods and demonstrate state-of-the-art FID results.

In the initial reviews, all reviewers commended our mathematical rigor and recognized the novelty of our unified framework. During the rebuttal phase, we provided detailed answers to all questions raised.

Particularly, we would like to emphasize an important point: most reviewers (e.g., e8ty, WUrW, wfTF) raised concerns regarding the orthogonality of the matrix A. As clearly stated in Section 3.3 and Appendix A of our manuscript, the orthogonality of A is a core component of our model and one of our principal contributions, which is also acknowledged by Reviewer wfTF.

Furthermore, we would like to restate that Reviewer wfTF, in the initial review (which occurred before the leakage issue) and again in the later response, explicitly stated their willingness of raising the score for our submission.

Once again, we sincerely thank you for your important role and for your efforts in preserving community trust and the integrity and fairness of the evaluation.

Respectfully,

Authors

---

### Meta-Review · Area_Chair_18pg · 2026-01-07

**Summary:**

Early scores were positive but a bit cautious, and most reviewers were leaning to accept. The main question was whether the orthogonality constraint is really needed. WUrW, e8ty, and wfTF asked if forcing the transform matrix A to be orthogonal would hurt expressivity. The rebuttal answers this point clearly. The authors show that without orthogonality the noise is no longer isotropic Gaussian, so the objective is no longer equivalent to the MAR and diffusion losses. The method also applies to earlier approaches such as VAR and xAR, adds little training cost under 3 percent, and reports strong results such as FID 1.15 on ImageNet-256, so AC recommends acceptance.

**Reviewer Concerns:**

Addressed:

Orthogonality: The rebuttal shows that a non-orthogonal A breaks isotropic Gaussian noise and the diffusion-loss equivalence, answering WUrW, e8ty, and wfTF.

Residual loss: Ablations with prefix, postfix, and random order support the residual-loss inductive bias, addressing wfTF.

Overhead and scaling: The authors report about 1 to 3 percent extra training cost and show longer-sequence results with the KL-8 tokenizer, addressing e8ty and WUrW.


Outstanding:

Reproducibility: Code is promised, but shV6 still lacks full logs and checkpoints, and the authors say they cannot release them for FFHQ due to policy.

**Reviewer Scores:**

shV6: Likely 4. Clarity is improved, but reproducibility is still partial.

e8ty: Likely 6. Overhead and matrix collapse concerns seem resolved.

WUrW: Likely 6. The orthogonality justification answers the core question.

wfTF: Likely 6 to 7. The reviewer promised to raise score.

---

### Decision · Program_Chairs · 2026-01-26

Accept (Poster)